# ABELIAN NEURAL NETWORKS

## ABSTRACT

In several domains such as natural language processing, it has been empirically reported that simple addition and subtraction in a somehow learned embedding space capture analogical relations. However, there is no guarantee that such relation holds for a new embedding space acquired by some training strategies. To tackle this issue, we propose to explicitly model analogical structure with an Abelian group. We construct an Abelian group network using invertible neural networks and show its universal approximation property. In experiments, our model successfully learns to capture word analogies from word2vec representations and shows better performance than other learning-based strategies. As a byproduct of modeling Abelian group operations, we furthermore obtain its natural extension to permutation invariant models with theoretical size-generalization capability.

## 1 INTRODUCTION

The vector representations of words called *word2vec* (Mikolov et al., 2013a;b) trained only on large unlabeled text data are known to capture linear regularities between words. For example, vec("king") − vec("man") + vec("woman") results in the most similar vector to vec("queen"). Similar results have been observed not only in other word embeddings (Mnih & Kavukcuoglu, 2013; Pennington et al., 2014) but also in various embedding spaces, such as combined embeddings of text and image (Kiros et al., 2014), emoji embeddings (Eisner et al., 2016), latent representation of deep convolutional generative adversarial networks (Radford et al., 2016), and feature space of pretrained image models (Upchurch et al., 2017).

Although the reports are interesting and attractive, such approaches have shortcomings. Since those methods of learning from unlabelled data usually do not explicitly incorporate learning analogical structure, there is no guarantee that the acquired embedding space has linear relations between instance pairs even if training itself works well. Even when an embedding space captures some kinds of analogies, it might not work for other kinds of analogies. Indeed, word2vec representation works well for inflectional analogies (68.22% accuracy) but poorly for encyclopedic analogies (7.11% accuracy) in our preliminary experiments (see Table 3 in Section 4.2). In such a case, it is quite difficult to tune the training algorithm for certain kinds of analogies you want to use.

To alleviate these issues, we propose to directly learn analogical relations on the embedding space from labeled data. One challenge in learning analogy in a supervised manner is how to model analogical functions. A naive way to do this is to train two separate models corresponding to addition and subtraction, respectively; however, it does not reflect the analogical structure and might be inefficient. In this work, we propose an *Abelian group network* to incorporate an analogical inductive bias into a neural network. The proposed network is designed to satisfy the Abelian group condition by using an invertible neural network. We also show that the Abelian group network is a universal approximator of smooth Abelian group operations. Since the inverse element in the Abelian group network and its gradient are analytically computable, we can train it for analogy tasks by common techniques for deep learning, such as stochastic gradient descent.

As a side effect of the algebraic structure, we can construct a permutation invariant function, i.e., a function for multisets, by repeatedly composing an Abelian group operation for multiple inputs. Multiset models can handle inputs of different sizes, and it is important for the models to automatically generalize between different size inputs, especially from small to large ones. However, existing multiset models (Zaheer et al., 2017; Qi et al., 2017) have no such theoretical guarantee. On

the other hand, our multiset models naturally induce the size-generalization capability because the output for larger inputs can be written as the composition of small elements. Further, we show that a necessary and sufficient condition for the composed function being permutation invariant is that the binary operation forms an Abelian semigroup, and we propose an *Abelian semigroup network*, by using the characterization of associative symmetric polynomials.

## 2 PRELIMINARIES

### 2.1 DEFINITIONS

In this section, let us introduce some basic notations and important definitions that will play a key role in this work.

#### 2.1.1 BASIC NOTATIONS

By $\mathbb{N}$, we represent the set of the natural numbers including $0$. We denote a vector by a bold symbol, e.g., $\boldsymbol{x}$. Let $\boldsymbol{x} \in \mathbb{R}^d$ be a $d$-dimensional vector. We represent the $i$-th element ($1 \leq i \leq d$) of $\boldsymbol{x}$ by $x_i$. For $1 \leq k \leq d$, $\boldsymbol{x}_{\leq k} \in \mathbb{R}^k$ is the $k$-dimensional vector $(x_1, \ldots x_k)$ and $\boldsymbol{x}_{<k} \in \mathbb{R}^{k-1}$ is the $(k-1)$-dimensional vector $(x_1, \ldots x_{k-1})$. We denote the elementwise product of two vectors $\boldsymbol{x}, \boldsymbol{y} \in \mathbb{R}^d$ by $\boldsymbol{x} \otimes \boldsymbol{y}$, such that $(\boldsymbol{x} \otimes \boldsymbol{y})_i = x_i y_i$ and the elementwise division of two vectors $\boldsymbol{x} \in \mathbb{R}^d, \boldsymbol{y} \in (\mathbb{R} \setminus \{0\})^d$ by $\boldsymbol{x} \oslash \boldsymbol{y}$, such that $(\boldsymbol{x} \oslash \boldsymbol{y})_i = x_i / y_i$. By $\| \cdot \|$, we represent the $L^2$ (Euclidean) norm.

#### 2.1.2 MULTISET AND PERMUTATION INVARIANCE

Here, we use $\mathcal{X}$ and $\mathcal{Y}$ to describe some domains, which are typically Euclidean spaces, i.e., $\mathcal{X} = \mathbb{R}^{d_1}$ and $\mathcal{Y} = \mathbb{R}^{d_2}$. We denote the set of multisets over $\mathcal{X}$ by $\mathbb{N}^{\mathcal{X}}$. We use $\{\boldsymbol{x}_1, \ldots \boldsymbol{x}_n\} \in \mathbb{N}^{\mathcal{X}}$ to describe a multiset composed of $\boldsymbol{x}_1, \ldots, \boldsymbol{x}_n \in \mathcal{X}$ (any confusion with sets is not problematic in this paper). The cardinality of a multiset is the number of elements with multiplicity and is expressed by $| \cdot |$, e.g., $|\{1, 2, 2, 3\}| = 4$.

For $n \in \mathbb{N}$, a symmetric group $S_n$ is the set of all $n!$ bijective functions $\sigma \colon \{1, 2, \ldots, n\} \to \{1, 2, \ldots, n\}$. For a permutation $\sigma \in S_n$ and $\boldsymbol{X} \in \mathcal{X}$, $\sigma \cdot X$ is defined such that $(\sigma \cdot \boldsymbol{X})_i = X_{\sigma(i)}$ for all $i \in \{1, \ldots, n\}$. A function $f \colon \mathcal{X}^n \to \mathcal{Y}$ is said to be permutation invariant if for any $\boldsymbol{X} \in \mathcal{X}^n$ and for any permutation $\sigma \in S_n$, $f(\sigma \cdot \boldsymbol{X}) = f(\boldsymbol{X})$ holds. This concept can be extended to functions that take vectors of different dimensions. Namely, a function $f \colon \bigcup_{k \in \mathbb{N}} \mathcal{X}^k \to \mathcal{Y}$ is called permutation invariant if for any $k \in \mathbb{N}$, for any $\boldsymbol{X} \in \mathcal{X}^k$ and for any permutation $\sigma \in S_k$, $f(\sigma \cdot \boldsymbol{X}) = f(\boldsymbol{X})$ holds. When $f \colon \bigcup_{k \in \mathbb{N}} \mathcal{X}^k \to \mathcal{Y}$ is permutation invariant, it can be also viewed as a function that takes multisets as input. For notation simplicity, we sometimes use the same symbol to express the multiset function: $f \colon \mathbb{N}^{\mathcal{X}} \to \mathcal{Y}$.

#### 2.1.3 UNIVERSALITY

Universality is an important theoretical property of neural networks' expressive power. Let $\mathcal{X}$ and $\mathcal{Y}$ be an input domain and an output domain, respectively. We consider a model $\mathcal{M}$ and a class of target functions $\mathcal{F}$, both of which are sets of functions $\mathcal{X} \to \mathcal{Y}$. The model $\mathcal{M}$ is a sup-universal approximator of $\mathcal{F}$ if for any target function $f^* \in \mathcal{F}$, for any $\epsilon > 0$, and for any compact subset $\mathcal{K} \subset \mathcal{X}$, there exists a function $f \in \mathcal{M}$ such that

$$\sup_{\boldsymbol{x} \in \mathcal{K}} \|f(\boldsymbol{x}) - f^*(\boldsymbol{x})\| < \epsilon. \tag{1}$$

If not noted otherwise, universality refers to the sup-universal property.

#### 2.1.4 BASIC ALGEBRA

Here, we introduce the basic definition of important algebraic structures in this study. Let $G$ be a set and $\circ \colon G \times G \to G$ be a binary operation. Below, we review four properties to define Abelian semigroups and groups.

**Associativity** For any $x, y, z \in G$, $(x \circ y) \circ z = x \circ (y \circ z)$.

**Identity Element** There exists an element $e \in G$, called the identity element, such that for any $x \in G$, $x \circ e = e \circ x = x$.

**Inverse Element** For any $x \in G$, there exists an element $x^{-1} \in G$, called the inverse element of $x$, such that $x \circ x^{-1} = x^{-1} \circ x = e$.

**Commutativity** For any $x, y \in G$, $x \circ y = y \circ x$.

Table 1 shows which properties are required in each algebraic structure. A semigroup only requires associativity to the binary operation. A group is a semigroup with an identity element and inverse elements. An Abelian (semi)group is a (semi)group with commutativity.

Table 1: Properties required for each algebraic structure.

| | Associativity | Identity | Inverse | Commutativity |
|---|---|---|---|---|
| Semigroup | ✓ | - | - | - |
| Group | ✓ | ✓ | ✓ | - |
| Abelian Semigroup | ✓ | - | - | ✓ |
| Abelian Group | ✓ | ✓ | ✓ | ✓ |

## 2.2 INVERTIBLE NEURAL NETWORKS

Invertible neural networks are neural networks that approximate invertible functions $\mathbb{R}^d \to \mathbb{R}^d$. Here, we review some existing studies for multi-dimensional case, i.e., $d \geq 2$, and single-dimensional case, i.e., $d = 1$.

### 2.2.1 NORMALIZING FLOWS

Multi-dimensional invertible neural networks have been studied mainly in the context of normalizing flows (Tabak & Vanden-Eijnden, 2010), which iteratively apply invertible functions to a simple original probability distribution to express complex probability distributions (Kobyzev et al., 2020; Papamakarios et al., 2019). There have been many variants proposed including residual flows (Behrmann et al., 2019), neural ODEs (Chen et al., 2018), and autoregressive flows (Kingma et al., 2017). Here we review affine coupling flows (Dinh et al., 2015), one of the most popular models with parallelizable efficient inverse computation. Each layer of the affine coupling flows maps $\boldsymbol{x} = (x_1, \ldots, x_d) \in \mathbb{R}^d$ to $\boldsymbol{y} = (y_1, \ldots, y_d) \in \mathbb{R}^d$ such that

$$\begin{cases} \boldsymbol{y}_{\leq k} = \boldsymbol{x}_{\leq k}, \\ \boldsymbol{y}_{>k} = \boldsymbol{x}_{>k} \otimes \exp(\alpha(\boldsymbol{x}_{\leq k})) + \beta(\boldsymbol{x}_{\leq k}), \end{cases} \tag{2}$$

where $\exp$ is applied elementwise and $\alpha, \beta \colon \mathbb{R}^k \to \mathbb{R}^{d-k}$ are trainable functions. The inverse is computed as follows:

$$\begin{cases} \boldsymbol{x}_{\leq k} = \boldsymbol{y}_{\leq k}, \\ \boldsymbol{x}_{>k} = (\boldsymbol{y}_{>k} - \beta(\boldsymbol{y}_{\leq k})) \otimes \exp(-\alpha(\boldsymbol{y}_{\leq k})). \end{cases} \tag{3}$$

They are used in many successful applications such as NICE (Dinh et al., 2015), Real NVP (Dinh et al., 2017), and Glow (Kingma & Dhariwal, 2018).

Although the normalizing flows have a limited form of transform, they still admit universalities on certain classes of functions (Teshima et al., 2020). The affine coupling flows are $L^p$-universal (weaker condition of universality) for $C^2$-diffeomorphisms. Some more complex models, including deep sigmoidal flows (Huang et al., 2018) and sum-of-squares polynomial flows (Jaini et al., 2019), are universal for $C^2$-diffeomorphisms.

### 2.2.2 SINGLE-DIMENSIONAL INVERTIBLE NEURAL NETWORKS

For single-dimensional functions, invertibility is equivalent to strict monotonicity. Monotonic networks (Sill, 1997) model strictly monotonic functions. The monotonically increasing version with

$K$ groups and $J_k$ units for $k$-th group is as follows:

$$f(x) = \min_{1 \le k \le K} \max_{1 \le j \le J_k} \exp(\tilde{w}^{(k,j)}) \cdot x + b^{(k,j)}, \tag{4}$$

where $\tilde{w}^{(k,j)}, b^{(k,j)} \in \mathbb{R}$ are trainable parameters. The monotonic networks are a universal approximator for strictly monotonic differentiable functions. Monotonic rational-quadratic transforms (Durkan et al., 2019) are another universal model for the single-dimensional case.

## 3 Proposed Methods

Here, we introduce the proposed methods. First, we propose a model for Abelian group operations and show its universality. Next, we explain how to model analogical relations by our model. Finally, we present architectures for multiset input and show the size-generalization ability.

### 3.1 Abelian Group Network

Let $\mathcal{X}$ be a Euclidean space, i.e., $\mathcal{X} = \mathbb{R}^d$ ($d \in \mathbb{N}$). We present the Abelian group network that models Abelian group operations as follows:

$$\boldsymbol{x} \circ \boldsymbol{y} = \phi^{-1}(\phi(\boldsymbol{x}) + \phi(\boldsymbol{y})), \tag{5}$$

where $\phi \colon \mathcal{X} \to \mathcal{X}$ is a trainable invertible function, typically modeled by an invertible neural network.

First, we check that this binary operation satisfies the four conditions of the Abelian group described in Section 2.1.4.

**Proposition 1** (Semigroup Conservation). *Let $\rho \colon \mathcal{X} \to \mathcal{X}$ be a bijective function. When $* \colon \mathcal{X} \times \mathcal{X} \to \mathcal{X}$ is associative, $\boldsymbol{x} \circ \boldsymbol{y} = \rho^{-1}(\rho(\boldsymbol{x}) * \rho(\boldsymbol{y}))$ is also associative. Similarly, when $*$ is commutative, $\circ$ is also commutative.*

*Proof.* Associativity:

$$
\begin{aligned}
(\boldsymbol{x} \circ \boldsymbol{y}) \circ \boldsymbol{z} &= \rho^{-1}(\rho(\rho^{-1}(\rho(\boldsymbol{x}) * \rho(\boldsymbol{y}))) * \rho(\boldsymbol{z})) \\
&= \rho^{-1}((\rho(\boldsymbol{x}) * \rho(\boldsymbol{y})) * \rho(\boldsymbol{z})) \\
&= \rho^{-1}(\rho(\boldsymbol{x}) * (\rho(\boldsymbol{y}) * \rho(\boldsymbol{z}))) \ (\because \text{Associativity of } *) \\
&= \rho^{-1}(\rho(\boldsymbol{x}) * \rho(\rho^{-1}(\rho(\boldsymbol{y}) * \rho(\boldsymbol{z})))) = \boldsymbol{x} \circ (\boldsymbol{y} \circ \boldsymbol{z}).
\end{aligned}
\tag{6}
$$

Commutativity:

$$
\begin{aligned}
\boldsymbol{y} \circ \boldsymbol{x} &= \rho^{-1}(\rho(\boldsymbol{y}) * \rho(\boldsymbol{x})) \\
&= \rho^{-1}(\rho(\boldsymbol{x}) * \rho(\boldsymbol{y})) \ (\because \text{Commutativity of } *) \\
&= \boldsymbol{x} \circ \boldsymbol{y}.
\end{aligned}
\tag{7}
$$

$\square$

By this proposition, since $+$ is associative and commutative, the Abelian group network is also associative and commutative. The identity element is

$$\boldsymbol{e} = \phi^{-1}(\boldsymbol{0}), \tag{8}$$

which satisfies

$$
\begin{aligned}
\boldsymbol{x} \circ \boldsymbol{e} &= \boldsymbol{x} \circ (\phi^{-1}(\boldsymbol{0})) \\
&= \phi^{-1}(\phi(\boldsymbol{x}) + \boldsymbol{0}) = \boldsymbol{x}
\end{aligned}
\tag{9}
$$

for all $\boldsymbol{x} \in \mathcal{X}$. The inverse element of $\boldsymbol{x} \in \mathcal{X}$ is

$$\boldsymbol{x}^{-1} = \phi^{-1}(-\phi(\boldsymbol{x})), \tag{10}$$

which satisfies

$$\begin{aligned}
\boldsymbol{x} \circ \boldsymbol{x}^{-1} &= \boldsymbol{x} \circ (\phi^{-1}(-\phi(\boldsymbol{x}))) \\
&= \phi^{-1}(\phi(\boldsymbol{x}) - \phi(\boldsymbol{x})) \\
&= \phi^{-1}(\boldsymbol{0}) = \boldsymbol{e}.
\end{aligned} \tag{11}$$

It is worth noting that we can analytically compute the inverse function (Equation 10).

Next, we check the expressive power of the Abelian group network. An Abelian Lie group is an Abelian group over a manifold in which the group operation $(x, y) \mapsto x \circ y$ and the inverse function $x \mapsto x^{-1}$ are both differentiable. The following theorem states that the Abelian group network can approximate any Abelian Lie group operation with arbitrary precision.

**Theorem 1** (Universality of Abelian group networks). *Let $\mathcal{X}$ be a Euclidean space. Abelian group networks are a universal approximator of Abelian Lie group operations over $\mathcal{X}$. In other words, for any Abelian Lie group operation $*: \mathcal{X} \times \mathcal{X} \to \mathcal{X}$, for any $\epsilon > 0$, and for any compact subset $\mathcal{K} \subset \mathcal{X}$, there exists a binary operation function $\circ: \mathcal{X} \times \mathcal{X} \to \mathcal{X}$ represented by an Abelian group network such that*

$$\sup_{\boldsymbol{x} \in \mathcal{K}, \boldsymbol{y} \in \mathcal{K}} \|(\boldsymbol{x} * \boldsymbol{y}) - (\boldsymbol{x} \circ \boldsymbol{y})\| < \epsilon. \tag{12}$$

We provide the proof in Appendix C.1. It is based on the theory of the Abelian Lie group and the universality of invertible neural networks.

## 3.2 Modeling Analogy by Abelian Group Network

Now we consider an analogical task of predicting an element $d$ in a relation $a : b = c : d$. Let $\boldsymbol{a}, \boldsymbol{b}, \boldsymbol{c}, \boldsymbol{d} \in \mathcal{X}$ be embedded vectors of corresponding elements. We need to define a model $f : \mathcal{X} \times \mathcal{X} \times \mathcal{X} \to \mathcal{X}$ that predicts $\boldsymbol{d}$ from $\boldsymbol{a}, \boldsymbol{b}$, and $\boldsymbol{c}$. Then, if the set of candidate elements are finite as in word embedding, we can search the nearest vector to $f(\boldsymbol{a}, \boldsymbol{b}, \boldsymbol{c})$. We explained in Section 1 that several studies heuristically use $f(\boldsymbol{a}, \boldsymbol{b}, \boldsymbol{c}) = \boldsymbol{b} - \boldsymbol{a} + \boldsymbol{c}$.

Analogical relations have two natural laws: when $a = b$, then $c = d$ holds; $a : b = c : d$ is equivalent to $a : c = b : d$. when $a = c$, then $b = d$ holds. Therefore, we would like to design $f$ to satisfy the two condition that $f(\boldsymbol{a}, \boldsymbol{b}, \boldsymbol{c}) = \boldsymbol{c}$ when $\boldsymbol{a} = \boldsymbol{b}$, and $f(\boldsymbol{a}, \boldsymbol{b}, \boldsymbol{c}) = f(\boldsymbol{a}, \boldsymbol{c}, \boldsymbol{b})$. These conditions hold in $f(\boldsymbol{a}, \boldsymbol{b}, \boldsymbol{c}) = \boldsymbol{b} - \boldsymbol{a} + \boldsymbol{c}$. However, when we try to parametrize $f$ by neural networks, it is not trivial to reflect these inductive biases using existing architectures. By using the Abelian group network, we can model $f$ so that it reflects the conditions as follows:

$$\begin{aligned}
f(\boldsymbol{a}, \boldsymbol{b}, \boldsymbol{c}) &= \boldsymbol{b} \circ \boldsymbol{a}^{-1} \circ \boldsymbol{c} \\
&= \phi^{-1}(\phi(\boldsymbol{b}) + \phi(\phi^{-1}(-\phi(\boldsymbol{a})))) \circ \boldsymbol{c}. \\
&= \phi^{-1}(\phi(\boldsymbol{b}) - \phi(\boldsymbol{a})) \circ \boldsymbol{c}. \\
&= \phi^{-1}(\phi(\boldsymbol{b}) - \phi(\boldsymbol{a}) + \phi(\boldsymbol{c})).
\end{aligned} \tag{13}$$

Also, this is strictly more expressive than the simple arithmetic operations because when $\phi$ is the identity, $\phi(\boldsymbol{a}, \boldsymbol{b}, \boldsymbol{c}) = \boldsymbol{b} - \boldsymbol{a} + \boldsymbol{c}$. We can train this model by minimizing the loss: $l(f(\boldsymbol{a}, \boldsymbol{b}, \boldsymbol{c}), \boldsymbol{d})$, where $l : \mathcal{X} \to \mathbb{R}$ can typically be an $L^2$ loss or negative cosine similarity.

## 3.3 Multiset Architecture and Size Generalization

Here, we construct a multiset architecture by combining the Abelian group networks and show it naturally generalizes in sizes. It is easy to check that if $\circ: \mathcal{X} \times \mathcal{X} \to \mathcal{X}$ is associative and commutative, $f: \mathcal{X}^k \to \mathcal{X}$ defined by

$$f(\boldsymbol{x}_1, \ldots, \boldsymbol{x}_k) = \boldsymbol{x}_1 \circ \boldsymbol{x}_2 \circ \cdots \circ \boldsymbol{x}_k \tag{14}$$

is permutation invariant. When this condition holds, we represent the multiset version of $f: \mathbb{N}^{\mathcal{X}} \to \mathcal{X}$ as follows by denoting a composition of $\circ$ by $\bigcirc$: $f(\boldsymbol{X}) = \bigcirc_{\boldsymbol{x} \in \boldsymbol{X}} \boldsymbol{x}$, where $\boldsymbol{X} \in \mathbb{N}^{\mathcal{X}}$ is a multiset of $\mathcal{X}$. By modeling $\circ$ by the Abelian group network, we can construct a multiset architecture of the Abelian group network:

$$f(\boldsymbol{X}) = \phi^{-1}\left(\sum_{\boldsymbol{x} \in \boldsymbol{X}} \phi(\boldsymbol{x})\right), \tag{15}$$

where the invertible function $\phi\colon \mathcal{X} \to \mathcal{X}$ is typically modeled by an invertible neural network.

Now, we consider the size-generalization ability of the multiset architectures. An intuitive explanation is as follows. As an extreme case, even if trained only on multisets of two elements, our models can learn the correct binary operation. Therefore, they generalize to multisets of larger sizes. In general, we can derive the following theorem. Appendix C.2 provides the proof.

**Theorem 2** (Size Generalization of Abelian Group Networks). *Let $f^*\colon \mathbb{N}^{\mathcal{X}} \to \mathcal{X}$ be a target function expressed by a composition of Abelian semigroup $(\mathcal{X}, \circ)$: $f^*(\boldsymbol{X}) = \bigcirc_{\boldsymbol{x} \in \boldsymbol{X}} \boldsymbol{x}$. Let $f\colon \mathbb{N}^{\mathcal{X}} \to \mathcal{X}$ be a multiset architecture of the Abelian group network: $f(\boldsymbol{X}) = \phi^{-1}\left(\sum_{\boldsymbol{x} \in \boldsymbol{X}} \phi(\boldsymbol{x})\right)$. When*

$$\|f(\boldsymbol{X}) - f^*(\boldsymbol{X})\| < \epsilon \tag{16}$$

*holds for any $\boldsymbol{X}$ ($\in \mathbb{N}^{\mathcal{X}}$) whose size is smaller than $a$ ($\geq 2$), then*

$$\|f(\boldsymbol{X}) - f^*(\boldsymbol{X})\| < \frac{\epsilon\left((aK_1K_2)^{\lceil \log_a b \rceil} - 1\right)}{aK_1K_2 - 1} \tag{17}$$

*holds for any $\boldsymbol{X}$ ($\in \mathbb{N}^{\mathcal{X}}$) whose size is $b$ ($\geq a$), under the condition that the Lipschitz constants of $\phi$ and $\phi^{-1}$ are $K_1$ and $K_2$, respectively.*

A necessary and sufficient condition of $f$ in Equation 14 being permutation invariant is that $\circ$ is an Abelian semigroup operation. In Appendix A, we provide the proof and propose an Abelian semigroup network by using the characterization of associative symmetric polynomials.

## 4 EXPERIMENTS

Here, we summarize the main experiment of learning word analogies. In addition to this section, we provide the experiments of size generalization on learning multiset functions in Appendix B.

### 4.1 COMMON SETTINGS

We implemented the neural networks in the PyTorch framework (Paszke et al., 2019) and optimized them using the Adam algorithm (Kingma & Ba, 2015). The hyperparameters for each model in each problem were tuned with validation datasets using the Bayesian optimization of the Optuna framework (Akiba et al., 2019). The experiments were run on Intel Xeon E5-2695 v4 with NVIDIA Tesla P100 GPU. See Appendix D for the detailed settings, such as the model architecture and the range of hyperparameters.

### 4.2 WORD ANALOGIES

In Section 1, We discussed the general motivation of training analogical functions over an embedding space. In addition, here, we explain the specific issue in word2vec (Mikolov et al., 2013a;b) and justify the motivation of using the Abelian group network. In word2vec, for predicting a word $d$ in a relation $a : b = c : d$, the word with the most similar vector to $\boldsymbol{b} - \boldsymbol{a} + \boldsymbol{c}$ (we denote the corresponding vector for each word by using a bold symbol) is selected in terms of the cosine similarity:

$$\cos(\boldsymbol{v}_1, \boldsymbol{v}_2) = \frac{\boldsymbol{v}_1 \cdot \boldsymbol{v}_2}{\|\boldsymbol{v}_1\|\|\boldsymbol{v}_2\|}. \tag{18}$$

Usually, the words $a, b, c$ are excluded from the candidate vocabulary under the assumption that a common word does not appear in one analogy example. Although this assumption is reasonable in many cases, it prevents us from solving certain problems such as a past tense verb analogy "do":"did" = "split":"split" or a plural noun analogy "apple":"apples" = "deer":"deer". On the other hand, if we do not exclude the words $a, b, c$ from the candidates, word2vec suffers from severe performance degradation e.g., falling from $73.59\%$ to $20.64\%$ in our preliminary experiment on the Google analogy test set. This is due to the nature of the word2vec algorithm: the result of the simple arithmetic calculation $\boldsymbol{b} - \boldsymbol{a} + \boldsymbol{c}$ has a high probability of being close to $\boldsymbol{b}$ or $\boldsymbol{c}$ in the cosine similarity, especially in a high dimensional space. We mitigate this issue by learning richer functions than addition and subtraction. In this experiment, we trained the Abelian group network from a labeled dataset and compared it with the original word2vec and other learning-based approaches.

**Word Embedding**   We used a 300-dimensional word2vec model for 3 billion words trained on Google News corpus of about 100 billion words[1]. We normalized each word embedding by $L^2$ norm, following the implementation of the Gensim framework (Řehůřek & Sojka, 2010).

**Word Analogy Models**   We compared different models for a word analogy function $f : \mathbb{R}^{300} \times \mathbb{R}^{300} \times \mathbb{R}^{300} \to \mathbb{R}^{300}$ that takes the vectors of words $a, b, c$ and predicts the vector of a word $d$. In the proposed method (WV + AGN), we modeled $f$ by Equation 13. For the invertible neural network $\phi : \mathbb{R}^{300} \to \mathbb{R}^{300}$, we implemented a model based on the affine coupling flows described in Section 2.2.1. In addition to the original word2vec (WV): $f(a, b, c) = b - a + c$, we prepared two trainable baselines based on a multilayer perceptron (WV + MLP and WV + MLP_C, respectively):

$$f(\boldsymbol{a}, \boldsymbol{b}, \boldsymbol{c}) = \text{MLP}(\boldsymbol{b} - \boldsymbol{a} + \boldsymbol{c}), \tag{19}$$

$$f(\boldsymbol{a}, \boldsymbol{b}, \boldsymbol{c}) = \text{MLP}_C(\text{CONCAT}(\boldsymbol{a}, \boldsymbol{b}, \boldsymbol{c})), \tag{20}$$

where we trained $\text{MLP} : \mathbb{R}^{300} \to \mathbb{R}^{300}$ or $\text{MLP}_C : \mathbb{R}^{900} \to \mathbb{R}^{300}$.

**Setup**   Except for WV, we trained $f$ by minimizing the loss function: $\text{loss}_f(\boldsymbol{a}, \boldsymbol{b}, \boldsymbol{c}, \boldsymbol{d}) = -\cos(f(\boldsymbol{a}, \boldsymbol{b}, \boldsymbol{c}), \boldsymbol{d})$ on the training set. We measured the accuracy on the test set by calculating the most similar vector to the model output for each word:

$$\underset{\boldsymbol{d} \in \mathcal{V}}{\arg\max} \cos(f(\boldsymbol{a}, \boldsymbol{b}, \boldsymbol{c}), \boldsymbol{d}), \tag{21}$$

where $\mathcal{V}$ is the set of all word embeddings in the word2vec model. For reference, we also tested the setting where we removed the words $a, b, c$ from the candidates:

$$\underset{\boldsymbol{d} \in \mathcal{V} \backslash \{\boldsymbol{a}, \boldsymbol{b}, \boldsymbol{c}\}}{\arg\max} \cos(f(\boldsymbol{a}, \boldsymbol{b}, \boldsymbol{c}), \boldsymbol{d}). \tag{22}$$

**Datasets**   We trained our models on the bigger analogy test set (Rogers et al., 2016), which consists of 4 categories, each of which has 10 smaller subcategories of 50 unique relations. First, we extracted the pairs included in the word2vec vocabularies. Then for each subcategory, randomly split them into a training set (60%), validation set (20%), and test set (20%). Finally, for each set, we generated all the combinations of the pairs for each subcategory and concatenated them among all subcategories. For some relations that contain multiple acceptable candidates, such as *mammal* and *canine* for hypernyms of *dog*, we used the first candidate for training and accepted any for the test. To check the transferability to another dataset, we also tested our models by the Google analogy test set (Mikolov et al., 2013a). It includes 19,544 question pairs (8,869 semantic and 10,675 syntactic), and all the words were included in the word2vec vocabulary. Tables 9 and 10 in Appendix D.2 summarize the explanation and the number of extracted pairs for all subcategories.

Table 2: Accuracy on bigger analogy test set when we selected from the whole words.

|                | num  | WV            | WV + MLP      | WV + MLP_C       | WV + AGN         |
| -------------- | ---- | ------------- | ------------- | ---------------- | ---------------- |
| Overall        | 3314 | 177 (5.34%)   | 565 (17.05%)  | 674 (20.34%)     | **690 (20.82%)** |
| Inflectional   | 900  | 100 (11.11%)  | 317 (35.22%)  | 340 (37.78%)     | **435 (48.33%)** |
| Derivational   | 882  | 4 (0.45%)     | 15 (1.70%)    | **40 (4.54%)**   | 20 (2.27%)       |
| Lexicographic  | 632  | 52 (8.23%)    | 154 (24.37%)  | **202 (31.96%)** | 172 (27.22%)     |
| Encyclopedic   | 900  | 21 (2.33%)    | 79 (8.78%)    | **92 (10.22%)**  | 63 (7.00%)       |

**Results**   In Table 2, we summarized the results on the bigger analogy test set when we selected from the whole vocabulary. In this setting, WV performed extremely poorly. WV + AGN achieved the best accuracy in overall, while the three trainable models all outperformed WV. Table 3 shows the accuracy comparison when we excluded $a, b, c$ from the candidates. While the proposed method outperformed WV again, interestingly, the other learning-based models degraded their performance compared to the original WV.

We summarized the results on the Google analogy test set in Tables 4 and 5 for each test setting. Here, MLP-based models significantly dropped performance while the proposed model did not.

---
[1]https://code.google.com/archive/p/word2vec/

Table 3: Accuracy on bigger analogy test set when we excluded $a, b, c$ from the candidates.

|  | num | WV | WV + MLP | WV + MLP_C | WV + AGN |
|---|---|---|---|---|---|
| Overall | 3314 | 864 (26.07%) | 569 (17.17%) | 686 (20.70%) | **1065 (32.14%)** |
| Inflectional | 900 | 614 (68.22%) | 324 (36.00%) | 369 (41.00%) | **656 (72.89%)** |
| Derivational | 882 | **103 (11.68%)** | 17 (1.93%) | 43 (4.88%) | 98 (11.11%) |
| Lexicographic | 632 | 83 (13.13%) | 151 (23.89%) | 188 (29.75%) | **205 (32.44%)** |
| Encyclopedic | 900 | 64 (7.11%) | 77 (8.56%) | 86 (9.56%) | **106 (11.78%)** |

Table 4: Accuracy of transfer test to Google analogy test set when we selected from the whole words.

|  | num | WV | WV + MLP | WV + MLP_C | WV + AGN |
|---|---|---|---|---|---|
| Overall | 19544 | 4033 (20.64%) | 1346 (6.89%) | 2222 (11.37%) | **5676 (29.04%)** |
| Semantic | 8869 | 1995 (22.49%) | 161 (1.82%) | 256 (2.89%) | **2260 (25.48%)** |
| Syntactic | 10675 | 2038 (19.09%) | 1185 (11.10%) | 1966 (18.42%) | **3416 (32.00%)** |

Only in the setting where we excluded $a, b, c$ from the candidates in the Google analogy test set, the proposed method did not outperform WV. This is possibly because the word2vec model was highly tuned for the Google analogy test set for this evaluation method. Indeed, it has been pointed out that word embedding algorithms are quite dependent on system design choices and hyperparameter tuning Levy et al. (2015). We show the full results of all subcategories on each dataset in each evaluation setting in Tables 11, 12, 13, and 14 in Appendix D.2.

Finally, we enumerated the answers of each model for some toy example relations in Table 6. The first four relations are the cases where $a = b$ or $a = c$. Thanks to the inductive bias incorporated in the Abelian group network, WV + AGN answered all the questions correctly as well as WV. WV + AGN also predicted the correct words for the last five relations, where we need to predict the words that appeared as $c$. On the other hand, the MLP-based models failed in most cases, which indicates that they are not good at the type of examples different from the training dataset.

Overall, while the naive learning approach overfitted to the certain dataset and evaluation criteria, the inductive biases incorporated in the Abelian group network successfully prevented the model from overfitting.

## 5 RELATED WORK

### 5.1 ALGEBRAIC STRUCTURES IN NEURAL NETWORKS

In the literature of deep learning, algebraic structures mainly appear in the context of group invariant/equivariant neural networks. For image input, some studies tried to incorporate reflection and rotation invariance into convolutional neural networks (Cohen & Welling, 2016; Worrall et al., 2017). Neural networks for (multi)sets (Zaheer et al., 2017; Qi et al., 2017) adopted invariance/equivariance to symmetric group actions. Recent studies have investigated symmetries invariant/equivariant to more general group actions, such as a subgroup of the symmetric group (Maron et al., 2019b) and sets of symmetric elements (Maron et al., 2020).

Our work differs from the above studies since we try to model an Abelian group/semigroup operation itself.

### 5.2 INDUCTIVE BIAS AND EXPRESSIVE POWER OF NEURAL NETWORKS

Inductive biases are assumptions on the nature of the data-generating process or the space of solutions in machine learning (Battaglia et al., 2018). Many studies have constructed special neural networks that reflect the inductive biases of a given problem setting. At the same time, since those networks are often composed of limited forms of neural operations, expressive power including universal approximation properties has been studied.

Table 5: Accuracy of transfer test to Google analogy test set when we excluded $a, b, c$ from the candidates.

|  | num | WV | WV + MLP | WV + MLP_C | WV + AGN |
|---|---|---|---|---|---|
| Overall | 19544 | **14382 (73.59%)** | 1427 (7.30%) | 2414 (12.35%) | 11857 (60.67%) |
| Semantic | 8869 | **6482 (73.09%)** | 163 (1.84%) | 256 (2.89%) | 4918 (55.45%) |
| Syntactic | 10675 | **7900 (74.00%)** | 1264 (11.84%) | 2158 (20.22%) | 6939 (65.00%) |

Table 6: Answers of each model for relations that include identical words.

|  | WV | WV + MLP | WV + MLP_C | WV + AGN |
|---|---|---|---|---|
| do:do = make:? | make | realize | ensuring | make |
| apple:apple = pen:? | pen | retractable_leash | dustcover | pen |
| do:did = do:? | did | failed | seemed | did |
| apple:apples = apple:? | apples | parsley_sprig | fruit | apples |
| do:did = split:? | split | failed | split | split |
| do:did = set:? | set | decided | established | set |
| do:did = put:? | put | allowed | Serge_Audate | put |
| apple:apples = deer:? | deer | raptor | fawn | deer |
| apple:apples = dice:? | dice | ethnic_heritages | Proximex_C### | dice |

Convolutional layers of convolutional neural networks (CNN) (LeCun et al., 1989) are designed to reflect spacial structures in images. CNN without fully connected layers has been shown to be universal (Zhou, 2018). Invertible neural networks incorporate the inductive bias of being bijective, which we summarized in Section 2.2. Message passing graph neural networks (Gilmer et al., 2017) such as graph convolutional networks (Kipf & Welling, 2017) and graph attention networks (Vaswani et al., 2017) are designed under the assumption that neighboring nodes have similar properties. They have been shown to have limited expressive power in terms of graph isomorphism (Xu et al., 2019; Morris et al., 2019) More expressive models have also been studied (Sato et al., 2019; Maron et al., 2019a; Keriven & Peyré, 2019; Maehara & Hoang, 2019). For a (multi)set learning problem, DeepSets (Zaheer et al., 2017) are one of the most popular models with universal approximation property.

## 5.3 SIZE GENERALIZATION

Graph neural networks and neural networks for (multi)sets can handle graphs and sets of different sizes, and their size-generalization ability has been empirically shown in some applications such as physical systems (Battaglia et al., 2016) and combinatorial optimization (Khalil et al., 2017; Abe et al., 2019; Veličković et al., 2020). However, from a theoretical perspective, there exist simple tasks on which graph neural networks do not naturally generalize to larger graphs (Yehudai et al., 2020). Recent work has analyzed the extrapolation of graph neural networks trained by gradient descent (Xu et al., 2021). There have been few studies on size generalization of (multi)sets probably because of difficulty in analyzing DeepSets for inputs of different sizes.

## 6 CONCLUSION AND FUTURE WORK

In this work, we presented a novel neural network architecture to model an Abelian group with universality. In the experiment of learning word analogies, we confirmed that the inductive bias of our model that reflects analogical structure successfully enhanced the performance. This brings up the possibility of solving other analogical tasks such as image generation by image analogies (Hertzmann et al., 2001), which is left as future work. Moreover, as an application other than analogies, we constructed a permutation invariant architecture (i.e., multiset model) by combining the Abelian group models, which has the theoretical capability of size generalization. We hope that our attempt to model algebraic structure by neural networks gives a new insight into the field of machine learning.

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

## A  ABELIAN SEMIGROUP NETWORK

### A.1  MOTIVATION

Below, we present a necessary and sufficient condition for multiset functions that are represented by the composition of binary operations to be well-defined.

**Proposition 2** (Permutation Invariant Conditions for Binary Operation)**.** *Let* $f : \bigcup_{k \in \mathbb{N}} \mathcal{X}^k \to \mathcal{X}$ *be a function represented as*

$$f(\boldsymbol{X}) = \boldsymbol{x}_1 \circ \cdots \circ \boldsymbol{x}_n, \tag{23}$$

*where* $\boldsymbol{X} = (\boldsymbol{x}_1, \ldots, \boldsymbol{x}_n) \in \mathcal{X}^n$ *and* $\circ : \mathcal{X} \times \mathcal{X} \to \mathcal{X}$ *is a binary operation (left-associative). The function* $f$ *is invariant if and only if* $\circ$ *forms an Abelian semigroup, namely,* $\circ$ *is commutative and associative.*

*Proof.* It is obvious that when $\circ$ is commutative and associative, $f$ is permutation invariant. Let us consider the case when $f$ is permutation invariant. $f((\boldsymbol{x}_1, \boldsymbol{x}_2)) = f((\boldsymbol{x}_2, \boldsymbol{x}_1))$ leads to $\boldsymbol{x}_1 \circ \boldsymbol{x}_2 = \boldsymbol{x}_2 \circ \boldsymbol{x}_1$ (commutativity). From $f((\boldsymbol{x}_1, \boldsymbol{x}_2, \boldsymbol{x}_3)) = f((\boldsymbol{x}_2, \boldsymbol{x}_3, \boldsymbol{x}_1))$, we have $(\boldsymbol{x}_1 \circ \boldsymbol{x}_2) \circ \boldsymbol{x}_3 = (\boldsymbol{x}_2 \circ \boldsymbol{x}_3) \circ \boldsymbol{x}_1$ and commutativity leads to $(\boldsymbol{x}_1 \circ \boldsymbol{x}_2) \circ \boldsymbol{x}_3 = \boldsymbol{x}_1 \circ (\boldsymbol{x}_2 \circ \boldsymbol{x}_3)$ (associativity). $\square$

On the basis of this proposition, our goal decomposes into learning Abelian semigroup operations over $\mathcal{X}$. In Section A.2, we propose neural network architectures for Abelian semigroups.

### A.2  ARCHITECTURE

Although the Abelian group network proposed in Section 3.1 is universal for smooth group operations, it is not sufficient for approximating an Abelian semigroup operation such as the product over $\mathbb{R}$, i.e., $x \circ y = xy$. Now we extend the Abelian group network and propose the Abelian semigroup network. Our idea is to extend $+$ of Equation 5 to a polynomial. From Proposition 1, Equation 5 is still a semigroup after we replace $+$ by a polynomial of $\boldsymbol{x}$ and $\boldsymbol{y}$ as long as the polynomial is associative and symmetric as a binary operation. We call the polynomials with this property *associative symmetric polynomials*, which are characterized by the following theorem. Since the original paper only gives a brief explanation, we give detailed proof in Appendix C.3.

**Theorem 3** (Characterization of Associative Symmetric Polynomials, Commutative Case of (Yoshida, 1963))**.** *An associative symmetric polynomial of* $x$ *and* $y$ *is one of the following three forms:*

$$x * y = \begin{cases} \alpha \\ \alpha + x + y \\ \frac{\beta(\beta-1)}{\gamma} + \beta(x + y) + \gamma xy & (\gamma \neq 0), \end{cases} \tag{24}$$

*where* $\alpha, \beta, \gamma$ *are coefficients.*

By applying this theorem to Proposition 1 for $\mathcal{X}$ with elementwise product and division, we obtain the following three kinds of Abelian semigroup operations:

$$\boldsymbol{x} \circ \boldsymbol{y} = \begin{cases} \rho^{-1}(\boldsymbol{\alpha}) & (25) \\ \rho^{-1}(\rho(\boldsymbol{x}) + \rho(\boldsymbol{y}) + \boldsymbol{\alpha}) & (26) \\ \rho^{-1}(\boldsymbol{\beta} \otimes (\boldsymbol{\beta} - \mathbf{1}) \oslash \boldsymbol{\gamma} + \boldsymbol{\beta} \otimes (\rho(\boldsymbol{x}) + \rho(\boldsymbol{y})) + \boldsymbol{\gamma} \otimes \rho(\boldsymbol{x}) \otimes \rho(\boldsymbol{y})), & (27) \end{cases}$$

where $\rho : \mathcal{X} \to \mathcal{X}$ is an invertible function and $\boldsymbol{\alpha}, \boldsymbol{\beta}, \boldsymbol{\gamma} \in \mathcal{X}$ are parameters ($\boldsymbol{\gamma}$ is nonzero for all elements in Equation 27). Equation 25 is a constant case, on which we do not put a focus due to its trivialness. Equation 26 forms a group where $\boldsymbol{e} = \rho^{-1}(-\boldsymbol{\alpha})$, $\boldsymbol{x}^{-1} = \rho^{-1}(-\rho(\boldsymbol{x}) - 2\boldsymbol{\alpha})$. It can be expressed by the Abelian group network with $\phi(\boldsymbol{x}) = \rho(\boldsymbol{x}) + \boldsymbol{\alpha}$ and $\phi^{-1}(\boldsymbol{x}) = \rho^{-1}(\boldsymbol{x} - \boldsymbol{\alpha})$ in Equation 5. Equation 27 is a semigroup but not a group. Just using this equation is fine, but we propose a simpler form, as the Abelian semigroup network:

$$\boldsymbol{x} \circ \boldsymbol{y} = \phi^{-1}(\phi(\boldsymbol{x}) \otimes \phi(\boldsymbol{y})), \tag{28}$$

where $\phi : \mathcal{X} \to \mathcal{X}$ is a trainable invertible function typically modeled by an invertible neural network. This is a special case of Equation 27 when $\boldsymbol{\beta} = \mathbf{0}, \boldsymbol{\gamma} = \mathbf{1}$ and therefore is a semigroup. Conversely, the Abelian semigroup network can express Equation 27 by

$$\phi(\boldsymbol{x}) = \boldsymbol{\gamma} \otimes \rho(\boldsymbol{x}) + \boldsymbol{\beta}, \ \phi^{-1}(\boldsymbol{x}) = \rho^{-1}((\boldsymbol{x} - \boldsymbol{\beta}) \oslash \boldsymbol{\gamma}). \tag{29}$$

Moreover, the Abelian semigroup network can approximate the Abelian group network: $\phi'^{-1}(\phi'(\boldsymbol{x}) + \phi'(\boldsymbol{y}))$. Let $\mathcal{X}_{>0} = \mathbb{R}_{>0}^d$, where $\mathcal{X} = \mathbb{R}^d$. One construction is approximating a bijective function $\pi : \mathcal{X} \to \mathcal{X}_{>0}$,

$$\pi(\boldsymbol{x}) = \exp(\phi'(\boldsymbol{x})), \; \pi^{-1}(\boldsymbol{x}) = \phi'^{-1}(\log(\boldsymbol{x})) \tag{30}$$

by $\phi$, where $\exp$ and $\log$ act elementwise. This is possible in any compact subset of $\mathcal{X}$. From the previous discussions so far, the Abelian semigroup network can approximate any binary operation which is homeomorphic to an associative symmetric polynomial. We confirm this fact in the experiments.

By calculating Equation 14, we can write the model for multiset input $\boldsymbol{X} = \{\boldsymbol{x}_1, \ldots \boldsymbol{x}_n\} \in \mathcal{X}$ in a simple form:

$$f(\boldsymbol{X}) = \phi^{-1}(\phi(\boldsymbol{x}_1) \otimes \cdots \otimes \phi(\boldsymbol{x}_n)), \tag{31}$$

where the invertible function $\phi : \mathcal{X} \to \mathcal{X}$ is typically modeled by an invertible neural network.

For the Abelian semigroup network, the error bound for small multiset propagates in the form of the product with the values $\phi(\boldsymbol{x}_i)$, which prevents us from inducing the bound like Theorem 2. However, it still has the size-generalization ability in most real applications where the values are not too large. We confirm this by an experiment in Section B.

## B  EXPERIMENTS OF SIZE GENERALIZATION

To check the size generalization over semigroup and group operations on multisets, we trained the models on synthetic data. The binary operation forms of the examined functions are $x \circ y = x + y, x + y + 1, \sqrt[3]{x^3 + y^3}$ (group cases) and $x \circ y = xy, x + y + \frac{xy}{2}$ (semigroup cases).

**Setup**  For the single-dimensional invertible neural network of the Abelian group network and Abelian semigroup network, we used monotonic networks (Sill, 1997). We tuned the hyperparameters, the number of groups and the number of units for each group. As a baseline, we used DeepSets (Zaheer et al., 2017), one of the most popular models for (multi)set learning. It incorporates two multilayer perceptrons (MLP). We used the same number of hidden layers for the two MLPs and tuned the hyperparameters, the number of layers in each MLP, the middle dimension, and the hidden dimension. Each model was trained to minimize the mean squared error on a training set.

**Data Generation**  As training data, we generated 500 multisets of size $\{2, 3, 4\}$ (chosen uniformly random). All the elements were single-dimensional and selected uniformly at random from $[-5.0, 5.0]$. A validation data of 100 multisets were generated from the same distribution. We prepared two kinds of test data. One consisted of 100 multisets drawn from the same distribution as the training and validation data, which we refer to by *small*. To see the size-generalization ability, the other consisted of 100 multisets of size $\{10, 11, 12\}$ (chosen uniformly at random) with the same element distribution, which we refer to by *large*.

**Results**  Table 7 summarizes the results. For the group functions, all models including the Abelian semigroup network performed well. This is consistent with the fact that group operations can be approximated by the Abelian semigroup network, as discussed in Section A. While the Abelian group network was better on the other two cases, DeepSets outperformed the Abelian group network on $\sqrt[3]{x^3 + y^3}$. This is possibly due to the optimization of MLPs in DeepSets being easier than monotonic networks in the Abelian group network and Abelian semigroup network. Invertible neural networks for the single-dimensional case that are easy to optimize are important for future work. For the semigroup operations, as well as DeepSets, the Abelian group network did not work well. This is reasonable because these semigroup operations can not be expressed by the Abelian group network.

On the size generalization, although DeepSets worked fairly well, our models worked better. For example, the Abelian semigroup network was better than DeepSets on *large* of $x + y$ despite being worse on *small*; The Abelian group network had similar results on $xy$ and $x + y + \frac{xy}{2}$.

Table 7: Mean squared error comparison between the models for each function. The upper three operations are groups and the lower two equations are semigroups. Square root of the values are presented. Smaller is better.

| $x \circ y$ | | | DeepSets ‖ | AGN | ASN |
|---|---|---|---|---|---|
| $x + y$ | small | | 0.00226 | **3.63e-7** | 0.0832 |
| | large | | 0.908 | **0.0366** | 0.309 |
| $x + y + 1$ | small | | 0.00772 | **4.17e-7** | 0.136 |
| | large | | 0.0335 | **0.0132** | 0.956 |
| $\sqrt[3]{x^3 + y^3}$ | small | | **0.0844** | 0.284 | 0.427 |
| | large | | **0.229** | 0.636 | 1.26 |
| $xy$ | small | | 13.0 | 36.7 | **0.00000295** |
| | large | | 28500 | 28390 | **31.5** |
| $x + y + \frac{xy}{2}$ | small | | 0.965 | 7.08 | **0.000660** |
| | large | | 194 | 193 | **1.22** |

## C    PROOFS

### C.1    PROOF OF THEOREM 1

First, we review a theorem of the Abelian Lie group, which is important in our proof. The real numbers $\mathbb{R}$ with the addition $+$ forms a Lie group, which we denote by $(\mathbb{R}, +)$. Also, the torus $\mathbb{T} = \mathbb{R}/2\pi\mathbb{Z}$ with the addition $+$ modulo $2\pi\mathbb{Z}$ forms a Lie group, which we denote by $(\mathbb{T}, +)$. It is known that any connected Abelian Lie group is isomorphic to $(\mathbb{R}, +)^k \times (\mathbb{T}, +)^h$ for some $k, h \in \mathbb{N}$ (Section 4.4.2 of (Procesi, 2007)).

Now, we give the proof of Theorem 1.

*Proof.* We use the fact that any connected Abelian Lie group is isomorphic to $(\mathbb{R}, +)^k \times (\mathbb{T}, +)^h$ for some $k, h \in \mathbb{N}$. Since there is no homeomorphic function $\mathbb{R}$ (which is not compact) to $\mathbb{T}$ (which is compact), Abelian Lie groups over a Euclidean space has the form of only $(\mathbb{R}, +)^k$. Therefore, any $*$ can be represented as

$$\boldsymbol{x} * \boldsymbol{y} = \pi^{-1}(\pi(\boldsymbol{x}) + \pi(\boldsymbol{y})), \tag{32}$$

where $\pi : \mathcal{X} \to \mathcal{X}$ is a homeomorphic function in terms of Lie groups, i.e., $\pi(\cdot)$ and $\pi^{-1}(\cdot)$ are analytic. Take any $\epsilon > 0$ and a compact subset $\mathcal{K} \subset \mathcal{X}$. We denote the image of $\mathcal{K} \times \mathcal{K}$ through the function $(\boldsymbol{x}, \boldsymbol{y}) \mapsto \pi(\boldsymbol{x}) + \pi(\boldsymbol{y})$ by $\mathcal{S}' = \{\pi(\boldsymbol{x}) + \pi(\boldsymbol{y}) \mid \boldsymbol{x}, \boldsymbol{y} \in \mathcal{K}\}$. Let

$$\mathcal{S} = \{\boldsymbol{s} \mid \exists \boldsymbol{s}' \in \mathcal{S}' \ s.t. \ \|\boldsymbol{s} - \boldsymbol{s}'\| \leq 2\epsilon\} \tag{33}$$

and

$$\mathcal{K}' = \mathcal{K} \cup \pi^{-1}(\mathcal{S}). \tag{34}$$

Then, we have a Lipschitz constant $L > 0$ of $\pi^{-1}$ over $\mathcal{S}$ since $\pi^{-1}$ is continuous and $\mathcal{S}$ is compact. Also, from the universality of invertible neural networks (Teshima et al., 2020) for the compact set $\mathcal{K}'$, there exists an invertible neural network $\phi : \mathcal{X} \to \mathcal{X}$ such that for any $\boldsymbol{x} \in \mathcal{K}'$

$$\|\pi(\boldsymbol{x}) - \phi(\boldsymbol{x})\| < \frac{\epsilon}{2L + 1} \tag{35}$$

and for any $\boldsymbol{x}' \in \pi(\mathcal{K}')$

$$\|\pi^{-1}(\boldsymbol{x}') - \phi^{-1}(\boldsymbol{x}')\| < \frac{\epsilon}{2L + 1}. \tag{36}$$

Then, for any $\boldsymbol{x}, \boldsymbol{y} \in \mathcal{K}$, $\phi(\boldsymbol{x}) + \phi(\boldsymbol{y}) \in \mathcal{S}(\subset \pi(\mathcal{K}'))$ because

$$\|(\pi(\boldsymbol{x}) + \pi(\boldsymbol{y})) - (\phi(\boldsymbol{x}) + \phi(\boldsymbol{y}))\| \leq \|\pi(\boldsymbol{x}) - \phi(\boldsymbol{x})\| + \|\pi(\boldsymbol{y}) + \phi(\boldsymbol{y})\|$$
$$< \frac{2\epsilon}{2L + 1} \tag{37}$$
$$< 2\epsilon.$$

Therefore, for any $\boldsymbol{x}, \boldsymbol{y} \in \mathcal{X}$, we have from the Lipshitz continuity of $\pi^{-1}$

$$
\begin{aligned}
\|\pi^{-1}(\pi(\boldsymbol{x}) + \pi(\boldsymbol{y})) - \pi^{-1}(\phi(\boldsymbol{x}) + \phi(\boldsymbol{y}))\| &\leq L\|(\pi(\boldsymbol{x}) + \pi(\boldsymbol{y})) - (\phi(\boldsymbol{x}) + \phi(\boldsymbol{y}))\| \\
&< L \cdot \frac{2\epsilon}{2L + 1} \\
&= \frac{2L\epsilon}{2L + 1}
\end{aligned}
\tag{38}
$$

and from Equation 36

$$
\|\pi^{-1}(\phi(\boldsymbol{x}) + \phi(\boldsymbol{y})) - \phi^{-1}(\phi(\boldsymbol{x}) + \phi(\boldsymbol{y}))\| < \frac{\epsilon}{2L + 1}.
\tag{39}
$$

From Equation 38 and 39, for any $\boldsymbol{x}, \boldsymbol{y} \in \mathcal{K}$, we obtain

$$
\begin{aligned}
\|(\boldsymbol{x} * \boldsymbol{y}) - (\boldsymbol{x} \circ \boldsymbol{y})\| &= \|\pi^{-1}(\pi(\boldsymbol{x}) + \pi(\boldsymbol{y})) - \phi^{-1}(\phi(\boldsymbol{x}) + \phi(\boldsymbol{y}))\| \\
&\leq \|\pi^{-1}(\pi(\boldsymbol{x}) + \pi(\boldsymbol{y})) - \pi^{-1}(\phi(\boldsymbol{x}) + \phi(\boldsymbol{y}))\| \\
&\quad + \|\pi^{-1}(\phi(\boldsymbol{x}) + \phi(\boldsymbol{y})) - \phi^{-1}(\phi(\boldsymbol{x}) + \phi(\boldsymbol{y}))\| \\
&\leq \frac{2L\epsilon}{2L + 1} + \frac{\epsilon}{2L + 1} \\
&= \epsilon.
\end{aligned}
\tag{40}
$$

This concludes that Abelian group networks are universal. $\qquad\square$

## C.2 PROOF OF THEOREM 2

*Proof.* We prove that for any $\boldsymbol{X} \in \mathbb{N}^{\mathcal{X}}$ of size smaller than $b \geq a$,

$$
\|f(\boldsymbol{X}) - f^*(\boldsymbol{X})\| < \frac{\epsilon\left((aK_1K_2)^{\lceil \log_a b \rceil} - 1\right)}{aK_1K_2 - 1}
\tag{41}
$$

by induction on size $b$. Note that $K_1K_2 > 1$ because they are the Lipschitz constants of inverse functions.

**Base Case** When $b = a$, Inequality 41 holds.

**Inductive Step** We assume Inequality 41 holds for size $b' = 1, \ldots, b - 1$. We divide $\boldsymbol{X}$ of size $b$ into balanced $a$ subsets $\boldsymbol{X_1}, \ldots, \boldsymbol{X_a}$ so that $\boldsymbol{X} = \boldsymbol{X_1} + \cdots \boldsymbol{X_a}$ and each $|\boldsymbol{X_i}| \leq \lceil \frac{b}{a} \rceil$ (Addition over multisets is defined as follows: $\{\boldsymbol{x}_1, \ldots, \boldsymbol{x}_n\} + \{\boldsymbol{x}_{n+1}, \ldots, \boldsymbol{x}_N\} = \{\boldsymbol{x}_1, \ldots, \boldsymbol{x}_n, \boldsymbol{x}_{n+1}, \ldots, \boldsymbol{x}_N\}$).

Then,

$$
\begin{aligned}
\|f(\boldsymbol{X}) - f^*(\boldsymbol{X})\| &= \left\| \phi^{-1}\left( \sum_{\boldsymbol{x} \in \boldsymbol{X}} \phi(\boldsymbol{x}) \right) - \bigcirc_{\boldsymbol{x} \in \boldsymbol{X}} \boldsymbol{x} \right\| \\
&= \left\| \phi^{-1}\left( \sum_{i=1}^{a} \sum_{\boldsymbol{x} \in \boldsymbol{X_i}} \phi(\boldsymbol{x}) \right) - \bigcirc_{i=1}^{a}\left( \bigcirc_{\boldsymbol{x} \in \boldsymbol{X_i}} \boldsymbol{x} \right) \right\| \\
&= \left\| \phi^{-1}\left( \sum_{i=1}^{a} \phi(f(\boldsymbol{X_i})) \right) - f^*\left( \{ f^*(\boldsymbol{X_1}), \ldots, f^*(\boldsymbol{X_a}) \} \right) \right\| \\
&\leq \left\| \phi^{-1}\left( \sum_{i=1}^{a} \phi(f(\boldsymbol{X_i})) \right) - \phi^{-1}\left( \sum_{i=1}^{a} \phi(f^*(\boldsymbol{X_i})) \right) \right\| + \\
&\qquad \left\| \phi^{-1}\left( \sum_{i=1}^{a} \phi(f^*(\boldsymbol{X_i})) \right) - f^*\left( \{ f^*(\boldsymbol{X_1}), \ldots, f^*(\boldsymbol{X_a}) \} \right) \right\| \\
&\leq K_2 \left\| \sum_{i=1}^{a} \phi(f(\boldsymbol{X_i})) - \phi(f^*(\boldsymbol{X_i})) \right\| + \\
&\qquad \| f\left( \{ f^*(\boldsymbol{X_1}), \ldots, f^*(\boldsymbol{X_a}) \} \right) - f^*\left( \{ f^*(\boldsymbol{X_1}), \ldots, f^*(\boldsymbol{X_a}) \} \right) \| \\
&< K_2 \left( \sum_{i=1}^{a} \| \phi(f(\boldsymbol{X_i})) - \phi(f^*(\boldsymbol{X_i})) \| \right) + \epsilon \\
&\leq K_2 \left( \sum_{i=1}^{a} K_1 \| f(\boldsymbol{X_i}) - f^*(\boldsymbol{X_i}) \| \right) + \epsilon \\
&= K_1 K_2 \left( \sum_{i=1}^{a} \| f(\boldsymbol{X_i}) - f^*(\boldsymbol{X_i}) \| \right) + \epsilon.
\end{aligned}
\tag{42}
$$

From the assumption on size $\lceil \frac{b}{a} \rceil$, we obtain

$$
\begin{aligned}
\|f(\boldsymbol{X}) - f^*(\boldsymbol{X})\| &< a K_1 K_2 \cdot \frac{\epsilon((aK_1 K_2)^{\lceil \log_a \lceil \frac{b}{a} \rceil \rceil} - 1)}{aK_1 K_2 - 1} + \epsilon \\
&< a K_1 K_2 \cdot \frac{\epsilon((aK_1 K_2)^{\lceil \log_a \frac{b}{a} \rceil} - 1)}{aK_1 K_2 - 1} + \epsilon \\
&< \frac{\epsilon\left((aK_1 K_2)^{\lceil \log_a b \rceil} - 1\right)}{aK_1 K_2 - 1},
\end{aligned}
\tag{43}
$$

which establishes the inductive step. $\qquad\square$

## C.3    PROOF OF THEOREM 3

*Proof.* First, we prove that associative polynomials are at most first-order for each variable. Assume that we have a $n$-order ($n \geq 2$) associative polynomial

$$
x * y = \sum_{i=0}^{n} \sum_{j=0}^{n} \alpha_{i,j} x^i y^j,
\tag{44}
$$

where $\alpha_{i,j} \in \mathbb{R}$ for $0 \leq i, j \leq n$. Then, we have

$$
(x * y) * z = \sum_{i=0}^{n} \sum_{j=0}^{n} \alpha_{i,j} (\sum_{k=0}^{n} \sum_{l=0}^{n} \alpha_{k,l} x^k y^l)^i z^j
\tag{45}
$$

and

$$
x * (y * z) = \sum_{i=0}^{n} \sum_{j=0}^{n} \alpha_{i,j} x^i (\sum_{k=0}^{n} \sum_{l=0}^{n} \alpha_{k,l} y^k z^l)^i.
\tag{46}
$$

Since $*$ is associative, these two must form an identity. By comparing a coefficient of $x^{n^2}$, we obtain

$$\sum_{j=0}^{n} \alpha_{n,j} (\sum_{l=0}^{n} \alpha_{n,l} y^l)^n z^j = 0. \tag{47}$$

If we have $0 \le j' \le n$ such that $\alpha_{n,j'} \ne 0$, from the coefficient of $z^{j'}$,

$$(\sum_{l=0}^{n} \alpha_{n,l} y^l)^n = 0. \tag{48}$$

Recursively, we get $\alpha_{n,0} = \alpha_{n,1} = \cdots = \alpha_{n,n} = 0$, which leads to contradiction. Therefore, we now have

$$\alpha_{n,0} = \alpha_{n,1} = \cdots = \alpha_{n,n} = 0. \tag{49}$$

In the same way, we can also prove

$$\alpha_{0,n} = \alpha_{1,n} = \cdots = \alpha_{n,n} = 0. \tag{50}$$

From Equation 49 and 50 for $n \ge 2$, now we know that associative polynomials are at most first-order for each variable. Therefore, symmetric associative polynomials have the form:

$$x * y = \alpha + \beta(x + y) + \gamma xy. \tag{51}$$

Then we have

$$(x * y) * z = \alpha + \beta((\alpha + \beta(x + y) + \gamma xy) + z) + \gamma(\alpha + \beta(x + y) + \gamma xy)z \tag{52}$$

and

$$x * (y * z) = \alpha + \beta((\alpha + \beta(x + y) + \gamma xy) + z) + \gamma(\alpha + \beta(x + y) + \gamma xy)z. \tag{53}$$

By solving this identity, we obtain

$$\alpha\gamma = \beta(\beta - 1). \tag{54}$$

This condition is equivalent to the associativity of $*$. It decomposes into three cases: $(\gamma = 0, \beta = 0)$, $(\gamma = 0, \beta = 1)$, and $\gamma \ne 0$. For each case, we obtain

$$x * y = \begin{cases} \alpha \\ \alpha + x + y \\ \frac{\beta(\beta-1)}{\gamma} + \beta(x + y) + \gamma xy \quad (\gamma \ne 0). \end{cases} \tag{55}$$

$\square$

# D EXPERIMENTAL DETAILS

Here, we explain the detailed setting and further discussion of the experiments that we did not cover in the main part.

## D.1 EXPERIMENTS OF SIZE GENERALIZATION

**Model Architecture** For the implementation of the monotonic networks, we followed the Equation 4, except that we added a coefficient term $s \in \mathbb{R}$ which automatically learn the sign of the weights:

$$f(x) = \min_{1 \le k \le K} \max_{1 \le j \le J_k} s \cdot \exp(\tilde{w}^{(k,j)}) \cdot x + b^{(k,j)}. \tag{56}$$

**Hyperparameters** All networks were trained by the Adam algorithm of $\mathrm{lr} = 10^{-3}$, beta $= (0.9, 0.999)$ for 1000 epochs with the batch size of 32. Hyperparameters of each model were tuned with the validation dataset using the Optuna framework for each function. For DeepSets, the number of layers for each MLP was selected from $[2, 8]$ and the middle dimension and hidden dimension were selected from $[2, 32]$. For the Abelian group network and Abelian semigroup network, the number of groups and the number of units in each group were selected from $[2, 32]$.

## D.2 WORD ANALOGIES

**Model architecture**  For the invertible neural network for the Abelian group network and Abelian semigroup network, we implemented a model based on the affine coupling flows using the FrEIA framework [2]. We stacked Glow coupling layers and random permutation layers of the dimensions in turn. For each Glow coupling layer, we used three layer feedforward neural networks with a hyperparameter of hidden_dim as a sub network. For MLP and MLP_C, we implemented multilayer perceptrons with the ReLU activation function.

**Hyperparameters**  All networks were trained by the Adam algorithm of $\mathrm{lr} = 10^{-3}$, $\mathtt{beta} = (0.9, 0.999)$ for 100 epochs with the batch size of 32. The hyperparameters of each model were tuned with the validation dataset using the Optuna framework. For MLP, the number of layers was selected from $[2, 6]$ and the hidden dimension was selected from $[8, 256]$. For the Abelian group network, the number of layers was selected from $[2, 6]$ and the hidden dimension was selected from $[8, 256]$. Weight_decay was selected from $[0, 10^{-3}]$ for all models.

Table 8 summarizes the selected hyperparameters for each model.

Table 8: Selected hyperparameters in word analogy task.

|             | layer_num | hidden_dim |
|-------------|-----------|------------|
| W2V         | -         | -          |
| W2V + MLP   | 4         | 223        |
| W2V + MLP_C | 2         | 516        |
| W2V + AGN   | 5         | 151        |

**Detailed Results**  In Table 9 and 10, we explained all the subcategories of the bigger analogy test set and the Google analogy test set. We summarized the full results for each subcategory in Table 11, 12, 13, and 14.

---

[2]https://github.com/VLL-HD/FrEIA

Table 9: Detailed explanation of bigger analogy test set. *pair* refers to the whole relation size, *used* refers to the number included in the word2vec model, and *identical* refers to the number of used relations that include identical words.

| category | subcategory | example | pair | used | identical |
|---|---|---|---|---|---|
| Inflectional | I01 noun - plural_reg | album:albums | 50 | 50 | 1 |
| Inflectional | I02 noun - plural_irreg | ability:abilities | 50 | 48 | 2 |
| Inflectional | I03 adj - comparative | angry:angrier | 50 | 49 | 0 |
| Inflectional | I04 adj - superlative | able:ablest | 50 | 49 | 0 |
| Inflectional | I05 verb_inf - 3pSg | accept:accepts | 50 | 50 | 1 |
| Inflectional | I06 verb_inf - Ving | achieve:achieving | 50 | 49 | 0 |
| Inflectional | I07 verb_inf - Ved | accept:accepted | 50 | 50 | 1 |
| Inflectional | I08 verb_Ving - 3pSg | adding:adds | 50 | 50 | 0 |
| Inflectional | I09 verb_Ving - Ved | adding:added | 50 | 50 | 0 |
| Inflectional | I10 verb_3pSg - Ved | adds:added | 50 | 50 | 0 |
| Derivational | D01 noun+less_reg | arm:armless | 50 | 48 | 0 |
| Derivational | D02 un+adj_reg | able:unable | 50 | 49 | 0 |
| Derivational | D03 adj+ly_reg | according:accordingl... | 50 | 49 | 0 |
| Derivational | D04 over+adj_reg | ambitious:overambiti... | 50 | 50 | 0 |
| Derivational | D05 adj+ness_reg | amazing:amazingness | 50 | 45 | 0 |
| Derivational | D06 re+verb_reg | acquire:reacquire | 50 | 48 | 0 |
| Derivational | D07 verb+able_reg | accept:acceptable | 50 | 49 | 0 |
| Derivational | D08 verb+er_irreg | achieve:achiever | 50 | 49 | 1 |
| Derivational | D09 verb+tion_irreg | accuse:accusation | 50 | 48 | 0 |
| Derivational | D10 verb+ment_irreg | accomplish:accomplis... | 50 | 47 | 0 |
| Encyclopedic | E01 country - capital | abuja:nigeria | 50 | 37 | 0 |
| Encyclopedic | E02 country - language | andorra:catalan | 50 | 36 | 0 |
| Encyclopedic | E03 UK_city - county | aberdeen:aberdeenshi... | 50 | 24 | 0 |
| Encyclopedic | E04 name - nationality | aristotle:greek | 50 | 23 | 0 |
| Encyclopedic | E05 name - occupation | andersen:writer/poet... | 50 | 27 | 0 |
| Encyclopedic | E06 animal - young | ape:baby/infant | 50 | 50 | 0 |
| Encyclopedic | E07 animal - sound | alpaca:bray | 50 | 50 | 0 |
| Encyclopedic | E08 animal - shelter | ant:anthill/insectar... | 50 | 50 | 0 |
| Encyclopedic | E09 things - color | ant:black/brown/red | 50 | 50 | 0 |
| Encyclopedic | E10 male - female | actor:actress | 50 | 48 | 0 |
| Lexicographic | L01 hypernyms - animals | allosaurus:dinosaur/... | 50 | 50 | 0 |
| Lexicographic | L02 hypernyms - misc | armchair:chair/seat/... | 50 | 50 | 0 |
| Lexicographic | L03 hyponyms - misc | backpack:daypack/kit... | 50 | 50 | 0 |
| Lexicographic | L04 meronyms - substance | atmosphere:gas/oxyge... | 50 | 50 | 1 |
| Lexicographic | L05 meronyms - member | acrobat:troupe | 50 | 50 | 0 |
| Lexicographic | L06 meronyms - part | academia:college/uni... | 50 | 47 | 4 |
| Lexicographic | L07 synonyms - intensity | afraid:terrified/hor... | 50 | 50 | 1 |
| Lexicographic | L08 synonyms - exact | airplane:aeroplane/p... | 50 | 50 | 0 |
| Lexicographic | L09 antonyms - gradable | able:unable/incapabl... | 50 | 50 | 0 |
| Lexicographic | L10 antonyms - binary | after:before/earlier... | 50 | 50 | 0 |

Table 10: Detailed explanation of Google analogy test set. num refers to the whole relation size and used refers to the number included in the word2vec model.

| category | subcategory | example | num | used |
|---|---|---|---|---|
| Semantic | capital-common-countries | Athens:Greece | 506 | 506 |
| Semantic | capital-world | Abuja:Nigeria | 4524 | 4524 |
| Semantic | currency | Algeria:dinar | 866 | 866 |
| Semantic | city-in-state | Chicago:Illinois | 2467 | 2467 |
| Semantic | family | boy:girl | 506 | 506 |
| Syntactic | gram1-adjective-to-adverb | amazing:amazingly | 992 | 992 |
| Syntactic | gram2-opposite | acceptable:unacceptable | 812 | 812 |
| Syntactic | gram3-comparative | bad:worse | 1332 | 1332 |
| Syntactic | gram4-superlative | bad:worst | 1122 | 1122 |
| Syntactic | gram5-present-participle | code:coding | 1056 | 1056 |
| Syntactic | gram6-nationality-adjective | Albania:Albanian | 1599 | 1599 |
| Syntactic | gram7-past-tense | dancing:danced | 1560 | 1560 |
| Syntactic | gram8-plural | banana:bananas | 1332 | 1332 |
| Syntactic | gram9-plural-verbs | decrease:decreases | 870 | 870 |

Table 11: Model comparison for each subcategory of bigger analogy test set.

| | num | WV | WV + MLP | WV + MLP_C | WV + AGN |
|---|---|---|---|---|---|
| I01 | 90 | 2(2.22%) | 5(5.56%) | **10**(**11.11**%) | 7(7.78%) |
| I02 | 90 | 0(0.00%) | 0(0.00%) | **8**(**8.89**%) | 0(0.00%) |
| I03 | 90 | 13(14.44%) | 22(24.44%) | 20(22.22%) | **41**(**45.56**%) |
| I04 | 90 | 10(11.11%) | 18(20.00%) | 25(27.78%) | **39**(**43.33**%) |
| I05 | 90 | 26(28.89%) | 58(64.44%) | 52(57.78%) | **60**(**66.67**%) |
| I06 | 90 | 14(15.56%) | 13(14.44%) | 19(21.11%) | **57**(**63.33**%) |
| I07 | 90 | 3(3.33%) | **54**(**60.00**%) | 42(46.67%) | 42(46.67%) |
| I08 | 90 | 11(12.22%) | 40(44.44%) | 51(56.67%) | **54**(**60.00**%) |
| I09 | 90 | 8(8.89%) | 49(54.44%) | 53(58.89%) | **62**(**68.89**%) |
| I10 | 90 | 13(14.44%) | 58(64.44%) | 60(66.67%) | **73**(**81.11**%) |
| D01 | 90 | **0**(**0.00**%) | **0**(**0.00**%) | **0**(**0.00**%) | **0**(**0.00**%) |
| D02 | 90 | 0(0.00%) | **1**(**1.11**%) | 0(0.00%) | 0(0.00%) |
| D03 | 90 | 1(1.11%) | 2(2.22%) | **9**(**10.00**%) | 5(5.56%) |
| D04 | 90 | **0**(**0.00**%) | **0**(**0.00**%) | **0**(**0.00**%) | **0**(**0.00**%) |
| D05 | 72 | 0(0.00%) | 2(2.78%) | **13**(**18.06**%) | 5(6.94%) |
| D06 | 90 | 0(0.00%) | 2(2.22%) | **8**(**8.89**%) | 0(0.00%) |
| D07 | 90 | **0**(**0.00**%) | **0**(**0.00**%) | **0**(**0.00**%) | **0**(**0.00**%) |
| D08 | 90 | 0(0.00%) | **4**(**4.44**%) | 1(1.11%) | 0(0.00%) |
| D09 | 90 | 3(3.33%) | 0(0.00%) | **8**(**8.89**%) | 7(7.78%) |
| D10 | 90 | 0(0.00%) | **4**(**4.44**%) | 1(1.11%) | 3(3.33%) |
| E01 | 56 | 0(0.00%) | 0(0.00%) | 1(1.79%) | **2**(**3.57**%) |
| E02 | 56 | 4(7.14%) | 14(25.00%) | **21**(**37.50**%) | 4(7.14%) |
| E03 | 20 | **6**(**30.00**%) | 0(0.00%) | 0(0.00%) | 3(15.00%) |
| E04 | 20 | 2(10.00%) | 3(15.00%) | 0(0.00%) | **4**(**20.00**%) |
| E05 | 30 | 5(16.67%) | 6(20.00%) | **8**(**26.67**%) | 5(16.67%) |
| E06 | 90 | 4(4.44%) | 36(40.00%) | **54**(**60.00**%) | 42(46.67%) |
| E07 | 90 | 3(3.33%) | **18**(**20.00**%) | 13(14.44%) | 7(7.78%) |
| E08 | 90 | 12(13.33%) | 39(43.33%) | 37(41.11%) | **56**(**62.22**%) |
| E09 | 90 | 10(11.11%) | 38(42.22%) | **61**(**67.78**%) | 35(38.89%) |
| E10 | 90 | 6(6.67%) | 0(0.00%) | 7(7.78%) | **14**(**15.56**%) |
| L01 | 90 | 0(0.00%) | 52(57.78%) | **55**(**61.11**%) | 38(42.22%) |
| L02 | 90 | 1(1.11%) | 15(16.67%) | **25**(**27.78**%) | 8(8.89%) |
| L03 | 90 | **0**(**0.00**%) | **0**(**0.00**%) | **0**(**0.00**%) | **0**(**0.00**%) |
| L04 | 90 | 0(0.00%) | 4(4.44%) | **10**(**11.11**%) | 4(4.44%) |
| L05 | 90 | 0(0.00%) | 1(1.11%) | **2**(**2.22**%) | 0(0.00%) |
| L06 | 90 | **9**(**10.00**%) | 0(0.00%) | 0(0.00%) | 6(6.67%) |
| L07 | 90 | **11**(**12.22**%) | 5(5.56%) | 0(0.00%) | 7(7.78%) |
| L08 | 90 | **0**(**0.00**%) | **0**(**0.00**%) | **0**(**0.00**%) | **0**(**0.00**%) |
| L09 | 90 | 0(0.00%) | **2**(**2.22**%) | 0(0.00%) | 0(0.00%) |
| L10 | 90 | **0**(**0.00**%) | **0**(**0.00**%) | **0**(**0.00**%) | **0**(**0.00**%) |

Table 12: Model comparison for each subcategory of bigger analogy test set.

| | num | WV | WV + MLP | WV + MLP_C | WV + AGN |
|---|---|---|---|---|---|
| I01 | 90 | 53(58.89%) | 5(5.56%) | 11(12.22%) | **55(61.11**%) |
| I02 | 90 | **42(46.67**%) | 0(0.00%) | 9(10.00%) | 30(33.33%) |
| I03 | 90 | **86(95.56**%) | 22(24.44%) | 20(22.22%) | 73(81.11%) |
| I04 | 90 | **68(75.56**%) | 18(20.00%) | 26(28.89%) | 64(71.11%) |
| I05 | 90 | **61(67.78**%) | 58(64.44%) | 55(61.11%) | **61(67.78**%) |
| I06 | 90 | 69(76.67%) | 13(14.44%) | 26(28.89%) | **71(78.89**%) |
| I07 | 90 | 52(57.78%) | 55(61.11%) | 42(46.67%) | **68(75.56**%) |
| I08 | 90 | 56(62.22%) | 42(46.67%) | 51(56.67%) | **69(76.67**%) |
| I09 | 90 | 58(64.44%) | 50(55.56%) | 63(70.00%) | **85(94.44**%) |
| I10 | 90 | 69(76.67%) | 61(67.78%) | 66(73.33%) | **80(88.89**%) |
| D01 | 90 | **0(0.00**%) | **0(0.00**%) | **0(0.00**%) | **0(0.00**%) |
| D02 | 90 | **3(3.33**%) | **3(3.33**%) | 0(0.00%) | 2(2.22%) |
| D03 | 90 | **26(28.89**%) | 2(2.22%) | 9(10.00%) | 15(16.67%) |
| D04 | 90 | **11(12.22**%) | 0(0.00%) | 0(0.00%) | 3(3.33%) |
| D05 | 72 | **21(29.17**%) | 2(2.78%) | 13(18.06%) | 17(23.61%) |
| D06 | 90 | 13(14.44%) | 2(2.22%) | 10(11.11%) | **19(21.11**%) |
| D07 | 90 | 1(1.11%) | 0(0.00%) | 0(0.00%) | **4(4.44**%) |
| D08 | 90 | 1(1.11%) | **4(4.44**%) | 1(1.11%) | 2(2.22%) |
| D09 | 90 | 21(23.33%) | 0(0.00%) | 9(10.00%) | **24(26.67**%) |
| D10 | 90 | 6(6.67%) | 4(4.44%) | 1(1.11%) | **12(13.33**%) |
| E01 | 56 | **18(32.14**%) | 0(0.00%) | 1(1.79%) | 5(8.93%) |
| E02 | 56 | 0(0.00%) | 14(25.00%) | **19(33.93**%) | 4(7.14%) |
| E03 | 20 | **0(0.00**%) | **0(0.00**%) | **0(0.00**%) | **0(0.00**%) |
| E04 | 20 | 0(0.00%) | 3(15.00%) | 0(0.00%) | **4(20.00**%) |
| E05 | 30 | 0(0.00%) | **6(20.00**%) | **6(20.00**%) | 2(6.67%) |
| E06 | 90 | 5(5.56%) | 33(36.67%) | **51(56.67**%) | 47(52.22%) |
| E07 | 90 | 3(3.33%) | 17(18.89%) | 12(13.33%) | **19(21.11**%) |
| E08 | 90 | 2(2.22%) | 40(44.44%) | 35(38.89%) | **53(58.89**%) |
| E09 | 90 | 12(13.33%) | 38(42.22%) | **57(63.33**%) | 33(36.67%) |
| E10 | 90 | **43(47.78**%) | 0(0.00%) | 7(7.78%) | 38(42.22%) |
| L01 | 90 | 7(7.78%) | 48(53.33%) | 50(55.56%) | **51(56.67**%) |
| L02 | 90 | 3(3.33%) | 15(16.67%) | **25(27.78**%) | 17(18.89%) |
| L03 | 90 | **3(3.33**%) | 0(0.00%) | 0(0.00%) | 2(2.22%) |
| L04 | 90 | 1(1.11%) | 3(3.33%) | **9(10.00**%) | 5(5.56%) |
| L05 | 90 | 1(1.11%) | 1(1.11%) | **2(2.22**%) | 1(1.11%) |
| L06 | 90 | 0(0.00%) | 0(0.00%) | 0(0.00%) | **1(1.11**%) |
| L07 | 90 | **12(13.33**%) | 4(4.44%) | 0(0.00%) | 2(2.22%) |
| L08 | 90 | **27(30.00**%) | 0(0.00%) | 0(0.00%) | 17(18.89%) |
| L09 | 90 | 2(2.22%) | **5(5.56**%) | 0(0.00%) | **5(5.56**%) |
| L10 | 90 | **8(8.89**%) | 1(1.11%) | 0(0.00%) | 5(5.56%) |

Table 13: Model comparison for each subcategory of Google analogy test set.

|  | num | WV | WV + MLP | WV + MLP_C | WV + AGN |
|---|---|---|---|---|---|
| capital-common-... | 506 | 225(44.47%) | 0(0.00%) | 0(0.00%) | **227**(**44.86**%) |
| capital-world | 4524 | 1168(25.82%) | 0(0.00%) | 0(0.00%) | **1223**(**27.03**%) |
| currency | 866 | **185**(**21.36**%) | 0(0.00%) | 0(0.00%) | 119(13.74%) |
| city-in-state | 2467 | 252(10.21%) | 0(0.00%) | 0(0.00%) | **350**(**14.19**%) |
| family | 506 | 165(32.61%) | 161(31.82%) | 256(50.59%) | **341**(**67.39**%) |
| gram1-adjective... | 992 | 15(1.51%) | 86(8.67%) | **156**(**15.73**%) | 84(8.47%) |
| gram2-opposite | 812 | 14(1.72%) | 200(24.63%) | **286**(**35.22**%) | 235(28.94%) |
| gram3-comparati... | 1332 | 329(24.70%) | 242(18.17%) | 433(32.51%) | **713**(**53.53**%) |
| gram4-superlati... | 1122 | 124(11.05%) | 244(21.75%) | 311(27.72%) | **406**(**36.19**%) |
| gram5-present-p... | 1056 | 73(6.91%) | 71(6.72%) | **219**(**20.74**%) | 160(15.15%) |
| gram6-nationali... | 1599 | **1180**(**73.80**%) | 0(0.00%) | 0(0.00%) | 996(62.29%) |
| gram7-past-tens... | 1560 | 134(8.59%) | 127(8.14%) | 252(16.15%) | **353**(**22.63**%) |
| gram8-plural | 1332 | 63(4.73%) | 82(6.16%) | 92(6.91%) | **176**(**13.21**%) |
| gram9-plural-ve... | 870 | 106(12.18%) | 133(15.29%) | 217(24.94%) | **293**(**33.68**%) |

Table 14: Model comparison for each subcategory of Google analogy test set.

|  | num | WV | WV + MLP | WV + MLP_C | WV + AGN |
|---|---|---|---|---|---|
| capital-common-... | 506 | **421**(**83.20**%) | 0(0.00%) | 0(0.00%) | 378(74.70%) |
| capital-world | 4524 | **3580**(**79.13**%) | 0(0.00%) | 0(0.00%) | 2689(59.44%) |
| currency | 866 | **304**(**35.10**%) | 0(0.00%) | 0(0.00%) | 180(20.79%) |
| city-in-state | 2467 | **1749**(**70.90**%) | 0(0.00%) | 0(0.00%) | 1223(49.57%) |
| family | 506 | 428(84.58%) | 163(32.21%) | 256(50.59%) | **448**(**88.54**%) |
| gram1-adjective... | 992 | **283**(**28.53**%) | 93(9.38%) | 162(16.33%) | **283**(**28.53**%) |
| gram2-opposite | 812 | 347(42.73%) | 201(24.75%) | 299(36.82%) | **419**(**51.60**%) |
| gram3-comparati... | 1332 | **1210**(**90.84**%) | 248(18.62%) | 437(32.81%) | 1044(78.38%) |
| gram4-superlati... | 1122 | **980**(**87.34**%) | 246(21.93%) | 319(28.43%) | 777(69.25%) |
| gram5-present-p... | 1056 | **825**(**78.12**%) | 78(7.39%) | 298(28.22%) | 729(69.03%) |
| gram6-nationali... | 1599 | **1438**(**89.93**%) | 0(0.00%) | 0(0.00%) | 1158(72.42%) |
| gram7-past-tens... | 1560 | 1029(65.96%) | 157(10.06%) | 283(18.14%) | **1061**(**68.01**%) |
| gram8-plural | 1332 | **1197**(**89.86**%) | 103(7.73%) | 122(9.16%) | 826(62.01%) |
| gram9-plural-ve... | 870 | 591(67.93%) | 138(15.86%) | 238(27.36%) | **642**(**73.79**%) |

