# OpenReview forum: "Abelian Neural Networks"
_ICLR.cc/2022/Conference — ICLR 2022 Submitted_

### Official Review · Reviewer_aHe9 · 2021-10-30

**Correctness:** 4
**Technical Novelty And Significance:** 3
**Empirical Novelty And Significance:** 3
**Recommendation:** 3
**Confidence:** 4

**Main Review:**

While the paper is generally reasonably clear, I feel that it takes a long time to get to the main idea, which is on page 5-6 of 9. The preliminaries (some of which are not actually used until the Appendix, I believe), related work, and elementary proofs on page 5, take up quite a lot of space.

So it's surprising to me that in the methods section, \phi, a critical component of the model, is left undefined. Its definition does not come until the bottom of page 7, in the experiments section, and even then, the definition is not an equation, but just the name of the network (Glow) and a citation.

Could you explain why commutativity is needed to model word analogies? Even if \circ is noncommutative, wouldn't you still have the property that a = b -> f(a,b,c) = c and a = c -> f(a,b,c) = b?

You write: “We mitigate this issue [of eliminating a, b, and c from the candidate set] by learning richer functions than addition and subtraction." But this seems to be not borne out by the experiments, where WV+AGN benefit greatly from eliminating a, b, and c from the candidate set, while WV+MLP and WV+MLP_C do not.

On the Google analogy test set, the proposed model does not outperform WV. You write: “This is probably because the word2vec model was highly tuned for the Google analogy test set for this evaluation method." While this is possible, do you have a citation or evidence for this claim?



**Summary Of The Paper:**

This paper introduces a new kind of neural network for multisets of vector inputs. Whereas DeepSets uses a function h(x, y) = g(f(x) + f(y)), the proposed model uses h(x, y) = f^{-1}(f(x) + f(y)), where f is an invertible neural network. It applies this network to the problem of learning word analogies.

**Summary Of The Review:**

This paper describes a simple and interesting method for modeling unordered data that has some nice theoretical properties, like generalization from smaller to larger sets. However, it is applied to only one problem, the word analogy problem, where the sets are always of size 3 and the importance of commutativity could be made more clear. Experimental results are mixed, with the proposed model performing the best on BATS but not on the Google analogies dataset.

---

> ### Author Response · Authors · 2021-11-15
> **Response to Reviewer aHe9**
>
> Thanks for your review and the insightful and important comments.
> Especially, Q3 is a critical point, which helped improve our paper a lot.
>
> ### C1: It takes a long time to get to the main idea.
> According to your comment, we have shortened the Preliminaries section and moved the Related Work section just before the Conclusion section in the final version.
> We kept the elementary proofs in the Proposed Method section to familiarize readers with our ideas.
> We do not believe that these matters of preference have a significant impact on the evaluation.
>
> ### C2: Definition of \phi.
> We have the definition of \phi just after it appeared in Section 4.1 (3.1 in the revised version): "where $\phi \colon \mathcal{X} \rightarrow \mathcal{X}$ is a trainable invertible function, typically modeled by an invertible neural network".
> Since we can use any invertible neural network architecture, the definition does not have to be an equation form.
> We also review the architecture of invertible neural networks in Section 2.2, including Glow architecture.
>
> ### Q3: Why is commutativity necessary to model analogies?
> We noticed from your comment that we were implicitly assuming another inductive bias of analogical relation: $a:b=c:d \Leftrightarrow a:c=b:d$.
> Therefore, $f$ needs to satisfy $f(\boldsymbol{a}, \boldsymbol{b}, \boldsymbol{c}) = f(\boldsymbol{a}, \boldsymbol{c}, \boldsymbol{b}) (=\boldsymbol{d})$.
> Then, by considering the case that $\boldsymbol{a}$ is the identity element in $\boldsymbol{b} \circ \boldsymbol{a}^{-1} \circ \boldsymbol{c} = \boldsymbol{c} \circ \boldsymbol{a}^{-1} \circ \boldsymbol{a}$, we obtain $\circ$ is commutative.
>
> ### C4: "This issue" is not born out by the experiments.
> We are afraid that you might be misinterpreting what we mean by "this issue" (We are sorry if not...).
> In the section on word analogies, we introduced the issue that occurs when **not** eliminating a, b, and c from the candidate set.
> It is borne out by the fact that all the trained models outperformed WV when we selected from the whole words.
>
> ### Q5: WV on the Google analogy test set.
> The results we observed (word embedding performing good on the Google analogy test set but poorly on another dataset) is consistent with existing work [1].
> Also, another paper claims that much of the performance gains of word embeddings are due to certain system design choices and hyperparameter optimizations [2].
>
> Please also note that we trained our models only on the bigger analogy test set, as described in the Datasets paragraph.
> Rather, we think it is amazing that the proposed model succeeded in zero-shot transfer while the other trainable models failed.
>
> ### Other comments
> Finally, we would like to respectfully claim that our contribution is not limited to a neural network for multiset input.
> Our main contribution is the design of a neural network for the Abelian group with universal approximation property, which you did not put much focus on in your comments.
>
> [1] Analogy-based Detection of Morphological and Semantic Relations With Word Embeddings: What Works and What Doesn’t. In SRW@HLT-NAACL, 2016.
>
> [2] Improving Distributional Similarity with Lessons Learned from Word Embeddings. Transactions of the Association for Computational Linguistics, 2015.

---

> > ### Comment · Reviewer_aHe9 · 2021-11-25
> > **Eliminating a, b, and c**
> >
> > I think the wording in my review was not clear; my understanding is that some methods perform well if you use the full candidate set including a, b, and c, and some methods perform poorly on the full candidate set, but better if you remove a, b, and c. The former is desirable, but it's well known that WV belongs to the latter category.
> >
> > My interpretation of the sentence "We mitigate this issue by learning richer functions" was to hope that the proposed richer functions place your method in the former category. That is, I hoped that the model with the richer functions would learn equally well from the full candidate set including a, b, and c, and the reduced candidate set excluding a, b, and c.
> >
> > However, this hope was not realized in tables 2 and 3: when excluding a, b, and c, WV improves from 5% to 26%, while WV+MLP and WV+NLP_C hold steady at 17% and 20%, respectively. But WV+AGN behaves like WV, improving from 21% to 32%. So while I grant that WV+AGN is the best system in both settings, WV+AGN clearly belongs to the latter category with WV, and it does not seem true that the richer functions used in AGN truly mitigate the issue (of needing to remove a, b, and c in order to achieve peak performance).
> >
> > Moreover, on the full candidate set including a, b, and c, WV+MLP_C performs the best in three out of four subtypes of analogies. It's only because WV+AGN does the best on the largest category that it still wins overall.

---

> > > ### Author Response · Authors · 2021-11-27
> > > **Re: Eliminating a, b, and c**
> > >
> > > Thanks for your additional explanation.
> > > It is natural that WV+AGN is closer to the latter category than MLP-based models because the invertibility constraint in AGN makes it closer to simple addition and subtraction than MLP.
> > > However, compared to WV, the performance degradation when selecting from the whole candidates is mitigated: while WV dropped by 20.73% (26.07% -> 5.34%), WV+AGN only dropped by 11.32% (32.14% -> 20.82%).
> > > On the Google analogy test set, while WV dropped by 53.55%, WV+AGN only dropped by 31.63%.
> > >
> > > We would also like to highlight the original motivations of explicitly modeling the analogies described in Section 1: give a guarantee that the models work and enable to train models for certain kinds of analogies.
> > > In terms of these motivations, the more practical setting is when we excluded a, b, and c from the candidates as Reviewer ezjt also pointed out.
> > > Only the accuracy in Table 2 might be considered as comparable, but considering the more practical setting (Table 3) and the robustness to other data (Tables 4 and 5), WV+AGN has a significant improvement from the MLP-based models.
> > > On the basis of your feedback, we will add these motivations and discussions in the experiment section of the final version.

---

### Official Review · Reviewer_ezJt · 2021-11-03

**Correctness:** 3
**Technical Novelty And Significance:** 3
**Empirical Novelty And Significance:** Not applicable
**Recommendation:** 6
**Confidence:** 3

**Main Review:**

### Pros
- Novel interesting idea of explicitly incorporating Abelian relation prior into neural networks.
- Theoretical proofs for the claimed properties regarding the AGN's ability to model Abelian relations and multisets.

### Cons
- Insufficient empirical evidence that AGN will help certain kinds of learning tasks in a realistic setting. The current evaluation is restricted to the word analogy task and does not show a definitive gain under practical settings. The strengths of the MLP baselines are questionable. Important details missing. See questions below and localized points below (Table 2-6).

I think the authors could improve the strength of the work by any of the following.
1) Looking for better tasks that can showcase the benefit of modeling Abelian relation explicitly. The arithmetic relation between word pairs is thought to be an emergent property of the popular word vector learning objectives. In other words, it may already be well-modeled by word vectors and thus AGN does not give a further advantage. At least a discussion about possible applications to a wide range of tasks would show the proposal has a greater impact.
2) Showing that AGN indeed gives a notable boost to word analogy in a practical setting. For example, not when WV is weakened when a, b, c are present (Table 2,4,6). Those words can easily be excluded in practice. I think Table 3 gives positive results but there are numbers in that table that seem off to me, which leads to the next point.
3) Making sure the experimental settings are described well and all the interesting/unusual observations from the results are addressed. For example, why would accuracy drop when excluding a, b, c, in Table 3? Why are MLPs so bad in Table 6? Those unusual observations may be related to problematic settings/processes, or unrealized behaviors/findings. In either case, it will be worth examining.
4) Comparing to competitive baselines. WV is an attested and competitive baseline. However, MLPs used in the paper seem to be particularly poor. Especially, why would they be poorer than WV (Table 3, 4, 5, 6) when they are built on top of WV? I think this suggests there is something wrong with the design and/or training. Thus, we don't really know if AGN has an advantage over MLP.
5) Showing AGN is advantageous in other aspects, e.g. interpretability, rather than focusing only on accuracy.

### Questions to the Authors
1) What is exactly the neural network architecture used in the experiments?
2) What are the authors' thoughts or plans in applying/experimenting AGN for other tasks / in other settings?
3) Localized points below for Table 2-6.

### Localized Points
1) Eq (5). Are w^{(k, j)}, b^{(k, j)} for 1 <= k <= K, 1 <= j <= J_k all trainable parameters?
2) End of P7. Worth explaining what is "the Glow architecture" used here.
3) Table 2. The competitive results for MLP_C suggest a simple linear underlying relation between embedding vectors. All the extra theory and design may be overkill.
4) Table 3. Compared to Table 2, for example, the last two numbers of MLP_C, I don't understand why there can be a drop? I think removing words a, b, c would only help? Are they cases where the correct answer is one of the words a, b, c, such as "split: split"? If that is the case, we should probably remove impossible queries or at least make a remark on the existence and frequency of such cases.
5) Table 4 and 5. The numbers make it seems like the benefit of AGN is to reduce confusion from a, b, c. However, we can easily exclude them in downstream applications.
6) After Table 5. "This is probably because the word2vec model was highly tuned for the Google analogy test set for this evaluation method." Do we have a better dataset, or even a better task, to showcase the benefit of the proposed approach?
7) Table 6, "we enumerated the answers of each model for some toy example relations". How are the examples chosen? Random? Or by some criteria?
8) Table 6, the first half. This may be a nice verification for the theoretical property, but it is hardly helpful in practice: one can write some simple logic to give the right answer for "a:a = b:?" queries.
9) Table 6, the second half. WV does just as perfect as AGN, so it does not particularly show an advantage. MLP and MLP_C failed completely. The comparison does not seem to support AGN is better, but rather there is something really bad happening with MLP and MLP_C and makes one doubt the legitimacy of them as competitive baselines.
10) After Table 6. "On the other hand, the MLP-based models failed in most cases, which indicates that they are not good at the type of examples different from the training dataset." I feel it impresses more as if we should put a better effort into designing and tuning baseline MLPs. Why are they giving nonsensical predictions while still making reasonable numbers in previous tables?

**Summary Of The Paper:**

The authors introduce Abelian group networks (AGN) that explicitly model the operational relation between elements in an Abelian group. The authors prove that the design has the ability to model such relations and present feasible neural network realizations. The authors also prove the ability of AGN to learn representations of multisets. The authors design experiments to test AGN's ability to model word analogy by viewing the "a:b = c:d" relation as $c - a + b$. The results seem to suggest that AGN outperforms MLP baselines always and word vectors in some cases. However, closer examinations reveal concerns about their strength and validity to support the efficacy of AGN.

**Summary Of The Review:**

Novel and interesting idea of adding explicit Abelian constraints into neural networks with proofs that AGN is able to model Abelian relations. However, the scope is limited to word analogy and experimental results do not provide good support that AGN is advantageous than word vectors and MLPs.

---

> ### Author Response · Authors · 2021-11-19
> **Response to Reviewer ezjt**
>
> Thanks for the many constructive comments and suggestions. (and a lot of commitment!!)
> Below, we address your questions and comments.
>
> ## Possible applications to other analogical tasks than word analogies.
> As explained in Section 1, the concept of analogy has been drawing attention recently in the area of machine learning.
> One of the most possible applications of our model other than word analogies is in image analogies [3].
> There have been some approaches to generate images using image analogies, by heuristically modeling analogical function [4] or in an unsupervised manner [5].
> On the other hand, our work gives a theoretical justification that the model satisfies the natural inductive biases in analogical relations and is applicable to any analogical relations.
> We mentioned this in Section 6 of the revised version.
>
> ## Concerns on word analogy experiments.
> ### Why does accuracy drop in Table 3?
> Thank you for pointing out this.
> The accuracy drop in Table 3 is caused by some relations that contain identical words, such as “series: series” in plural analogy, which we did not realize are included in the bigger analogy test set.
> The proportion of these examples are quite small and we think it does not affect the overall evaluation.
> We listed the number of such cases for each category in Table 9 in Appendix D.2.
>
> ### Are MLP baselines legitimate?
> We are sure that the baselines are legitimate and the comparison is fair.
> Please note that the experiment on the Google analogy test set is a transfer test.
> All the learning-based models are tuned by the accuracy on the validation set of the bigger analogy test in the setting that we select from the whole vocabulary (tuned hyperparameters are described in Appendix D.2).
> The competitive performance of MLP_C with AGN in this setting indicates that tuning of these models is fair enough.
> Since the example in the Google analogy test set is a zero-shot transfer and we can not see the accuracy during tuning time, the drop in performance of MLP-based models is not due to something like lack of tuning.
> Rather, we think it is amazing that our model still works well in the transfer dataset and we consider it well demonstrates that the proposed algebraic architecture enhanced the model's robustness to other kinds of analogies that are not included in training data.
> Our hypothesis for the cause of this is as follows.
> While MLPs tend to focus only on some part of the information required for analogies included in training data and omit the other information, the invertible neural network in AGN keeps all the information included in the embeddings (because it is invertible) and can extrapolate some analogies not included in the training data.
>
> I hope we have addressed your concerns about the word analogy experiments so far, but please clarify if you still have any concerns.
>
> ## Questions
> ### Q1
> We describe the detailed neural network architectures in Appendix D.2 including selected hyperparameters by hyperparameter search.
> ### Q2
> Answered above.
>
> ## Localized Points
> ### 1
> Yes, they are all trainable parameters. We added an explanation in the revision.
> ### 2
> We added a reference to Section 2.2.1, where we review the architecture of the affine coupling flows.
> We deleted the word “Glow” from here, since it is only implementation-specific and described in Appendix D.2.
> ### 3
> Please note that AGN gives a constraint on the form of a function, while MLP_C can express more complex functions.
> As you also pointed out, the most practical setting is in Table 3, where AGN significantly outperformed the baselines.
> ### 4
> Answered above.
> ### 5
> The experiments on Tables 4 and 5 are not aimed at real-world applications, but to measure the robustness of trained models for other datasets than they are trained on.
> ### 6
> We could not find another word analogy dataset.
> Also, we added a reference that supports our claim that word2vec might be highly tuned for the Google analogy test set. (Please also see Q5 to Reviewer aHe9 for this point.)
> ### 7
> The examples were chosen at random under the condition that they were not included in the training data.
> ### 8
> Your point is right and the results of AGN in Table 6 are just a theoretical verification.
> ### 9, 10
> We addressed your concerns on the validity of the experiment above.
> While Tables 4 and 5 quantitatively demonstrate that learning-based models except for AGN are not robust to analogies that are not included in training data, Table 6 qualitatively demonstrates that by examples.
>
> [3] Image analogies. Proceedings of the 28th annual conference on Computer graphics and interactive techniques, 2001.
>
> [4] Deep Visual Analogy-Making. NIPS, 2015.
>
> [5] Visual attribute transfer through deep image analogy. ACM Transactions on Graphics, 2017.

---

> > ### Comment · Reviewer_ezJt · 2021-12-04
> > **Response to the Authors**
> >
> > Thanks for the response! It helped clarify many of my points. However, the main weakness remains -- weak experimental results on one task of limited impact. The mathematically well-motivated idea of incorporating Abelian constraints into NNs will likely inspire the community. I personally believe so, but there is not yet strong evidence that it provides an advantage in practical settings. Thus, I will keep the weak accept recommendation. I think if the authors can revise upon the many good suggestions here, it will become a much stronger submission.

---

### Official Review · Reviewer_uoK8 · 2021-11-08

**Correctness:** 3
**Technical Novelty And Significance:** 3
**Empirical Novelty And Significance:** 2
**Recommendation:** 6
**Confidence:** 4

**Details Of Ethics Concerns:**

Does not have any ethical concerns in my view.

**Main Review:**

Strengths:

1. The approach is motivated well, the proposed abelian network is very interesting and could have applications for other tasks.
2. The paper is written well and is very easy to follow

Weaknesses:

The experimental setup seems quite weak:
1. The authors only consider only word embedding setup (word2vec) which is several years old and many other better-performing embeddings (on word analogy task) have since been proposed such as fasttext or glove.
2. Even with word2vec, the proposed approach only outperforms baselines in a specific case (where we exclude query words from the candidate set).
3. Why is this problem interesting in itself? A side effect of learning unsupervised word embeddings was they perform well on word analogies, however, it is not clear why training of this specific task is important. Does that improve the embeddings for any downstream tasks where they can be used?

----

Updated the score from 5 to 6 after rebuttal


**Summary Of The Paper:**

This paper presents a method of learning word analogies by training a network to explicitly model analogical relationships using an architecture based on invertible projections, which they show induces an abelian group on the input space. With a supervised setup using a training set of word analogies, the paper shows improvements over baselines that do not model this structure.

**Summary Of The Review:**

While the presented premise and the theory of abelian neural networks is quite interesting and may have interesting applications, the experimental section seems weak and needs additional results/motivation.

---

> ### Author Response · Authors · 2021-11-18
> **Response to Reviewer uoK8**
>
> Thanks for your review and insightful comments.
> We are glad that you find our approach interesting.
> Below, we address your questions and comments.
> ### C1: Word2vec is old.
> The goal of our word analogy experiments is not like obtaining SOTA performance but rather verifying our model's capability of modeling analogical relations.
> We considered any word embedding is fine for this purpose and chose the most famous one.
>
> ### C2: The proposed approach outperforms baselines in a specific case.
> We are afraid that you might be missing that we trained the models on the bigger analogy test set, as described in the Datasets paragraph.
> On the test data of this dataset, the proposed model outperformed the baselines in both cases where we do and do not exclude query words from the candidate set.
> In the transferability test to the Google analogy test set, we think it is rather amazing that the proposed model succeeded in zero-shot transfer, unlike the other learning-based models, and outperformed WV, which possibly overfitted to this data, when we excluded query words from the candidate.
>
> ### Q3: Is modeling word analogies important?
> We conducted the word analogy experiment to show that our model is suitable for modeling analogical relations, but the application is not limited to this task.
> As explained in Section 1, recently analogical structures have been appearing in many domains such as natural language processing and images.
> We believe our fundamental model has a great potential to be used for a variety of analogical tasks such as image generation.

---

> > ### Comment · Reviewer_uoK8 · 2021-11-22
> > **Re: Response to Reviewer uoK8**
> >
> > Thank you to the authors for their detailed response and clarifications!
> >
> > > The goal of our word analogy experiments is not like obtaining SOTA performance
> >
> > I agree that this is not the goal, however, since current experiments only consider one kind of embeddings, it is not clear that this method applied on top of any embedding methods will give improvements on a word analogy task. The model's capability of modeling analogical relations should at least be better than word embedding methods that do not explicitly model it. I think this work could still benefit from more experiments.
> >
> > > Even with word2vec, the proposed approach only outperforms baselines in a specific case
> >
> > I mistyped "exclude" instead of "include" in my original review. But thank you for clarifying this and explaining the experiments in more detail. I have improved my score to 6.
> >
> > > We believe our fundamental model has a great potential to be used for a variety of analogical tasks such as image generation.
> >
> > This would make a much stronger paper, if this potential was explored for at least one more task

---

> > > ### Author Response · Authors · 2021-11-27
> > > **Thank you for the score update!**
> > >
> > > We appreciate your additional comments.
> > > In this work, we focused on constructing the fundamental model and its theory and demonstrated it works in a simple analogy task.
> > > We will leave your additional suggestion of more complex applications as important future work.

---

### Author Response · Authors · 2021-11-30
**Summary**

We deeply appreciate the insightful questions and comments from the reviewers.
While there was agreement that our idea is interesting and the theories are solid, the main concerns of the reviewers were on the superiority of our model in the experiments of word analogy.
We highlighted the fact that the experiment on the Google analogy test set is a zero-shot transfer test, which many reviewers seemed to be missing.
Taking this into account, our model has a clear improvement in accuracy from other learning-based models both in the practical setting and in terms of robustness to other analogies that are not included in training data.
Also, we addressed other comments in individual replies.

---

### Decision · Program_Chairs · 2022-01-20

**Decision:**

Reject

**Comment:**

The authors consider an interesting approach for modeling analogical relations through Abelian group networks. While the conceptual contributions in the work, the explicit introduction of Abelian relations in particular, were generally appreciated, the reviewers found the numerical results provided in the paper lacking. In addition, several issues regarding the scope of the problems to which the proposed approach applies have been raised. Thus, given this, and the exchanges between the reviewers and the authors, in its present form, the paper cannot be recommended for acceptance. The authors are encouraged to incorporate the valuable feedback provided by the knowledgeable reviewers.